# An unfolded protein-induced conformational switch activates mammalian IRE1

G Elif Karagöz[1]*, Diego Acosta-Alvear[1†], Hieu T Nguyen[2], Crystal P Lee[1‡], Feixia Chu[2], Peter Walter[1]*

[1]Department of Biochemistry and Biophysics, Howard Hughes Medical Institute, University of California, San Francisco, San Francisco, United States; [2]Department of Molecular, Cellular, and Biomedical Sciences, University of New Hampshire, Durham, United States

**Abstract** The unfolded protein response (UPR) adjusts the cell's protein folding capacity in the endoplasmic reticulum (ER) according to need. IRE1 is the most conserved UPR sensor in eukaryotic cells. It has remained controversial, however, whether mammalian and yeast IRE1 use a common mechanism for ER stress sensing. Here, we show that similar to yeast, human IRE1α's ER-lumenal domain (hIRE1α LD) binds peptides with a characteristic amino acid bias. Peptides and unfolded proteins bind to hIRE1α LD's MHC-like groove and induce allosteric changes that lead to its oligomerization. Mutation of a hydrophobic patch at the oligomerization interface decoupled peptide binding to hIRE1α LD from its oligomerization, yet retained peptide-induced allosteric coupling within the domain. Importantly, impairing oligomerization of hIRE1α LD abolished IRE1's activity in living cells. Our results provide evidence for a unifying mechanism of IRE1 activation that relies on unfolded protein binding-induced oligomerization.

DOI: https://doi.org/10.7554/eLife.30700.001

*For correspondence:
elif@walterlab.ucsf.edu (GEK);
peter@walterlab.ucsf.edu (PW)

Present address: †Department of Molecular, Cellular and Developmental Biology, University of California, Santa Barbara, Santa Barbara, United States; ‡BioMarin Pharmaceutical Inc, San Rafael, United States

**Competing interests:** The authors declare that no competing interests exist.

## Introduction

Protein-folding homeostasis is critical for proper cell function. Accordingly, cells evolved surveillance mechanisms to monitor protein-folding status and elicit adaptive responses to adjust protein-folding capacity according to need (*Balchin et al., 2016*; *Bukau et al., 2006*; *Walter and Ron, 2011*). In the endoplasmic reticulum (ER), where the majority of transmembrane and soluble secretory proteins fold and mature, protein-folding homeostasis is ensured by a network of signaling pathways collectively known as the unfolded protein response (UPR) (*Walter and Ron, 2011*). In metazoans, perturbations leading to the accumulation of mis- or unfolded proteins in the ER are recognized as 'ER stress' by three unique ER-resident UPR sensors, IRE1, PERK and ATF6 (*Cox et al., 1993*; *Cox and Walter, 1996*; *Harding et al., 2000*; *Niwa et al., 1999*; *Sidrauski and Walter, 1997*; *Tirasophon et al., 2000*; *Walter and Ron, 2011*; *Yoshida et al., 1998*; *Yoshida et al., 2001*). These sensors transmit information about the protein-folding status in the ER and drive gene expression programs that modulate both the protein-folding load and folding capacity of the ER. If ER stress remains unmitigated, the UPR induces pro-apoptotic pathways, thereby placing the network at the center life-or-death decisions that affect the progression of numerous diseases (*Bi et al., 2005*; *Feldman et al., 2005*; *Lin et al., 2007*; *Lu et al., 2014*; *Vidal et al., 2012*; *Walter and Ron, 2011*; *Zhang and Kaufman, 2008*).

IRE1 drives the most conserved branch of the UPR, which exhibits remarkably similar mechanistic aspects shared between yeast and mammals (*Aragón et al., 2009*; *Korennykh et al., 2009*; *Li et al., 2010*). In mammals, IRE1 exists in two isoforms, α and β. IRE1α is ubiquitously expressed,

**eLife digest** Proteins are long string-like molecules that fold into specific three-dimensional shapes. Most proteins that a cell uses to communicate with its environment are folded within a part of the cell called the endoplasmic reticulum. Dedicated sensor proteins in this cellular compartment track this process to make sure that it continues to meet the cell's demand for protein folding. If it cannot meet the demand, unfolded or poorly folded proteins build up, which stresses the cell.

IRE1 is a sensor protein that detects stress in the endoplasmic reticulum. It is found in a range of organisms from yeast to humans, where it spans the membrane that encloses the endoplasmic reticulum. When unfolded proteins accumulate, IRE1 proteins come together and form so-called oligomers. The IRE1 oligomers then become active and send signals outside of the endoplasmic reticulum. These signals adjust the cell's protein-folding capacity according to its needs at that time.

The yeast version of IRE1 directly recognizes unfolded proteins in the endoplasmic reticulum. Yet, its human counterpart was found to have a different three-dimensional structure, which suggested that it might use a different mechanism to detect the stress.

Now, Karagöz et al. show that, as in yeast, the sensor part of human IRE1 does indeed bind to unfolded proteins directly. This binding causes this part of the protein to engage other copies of IRE1 and form the oligomers. To understand this interaction in more detail, Karagöz et al. used a technique called nuclear magnetic resonance spectroscopy to monitor changes in the shape of proteins. These observations revealed that binding to an unfolded protein causes other parts of IRE1 protein to change shape. In turn, these shape changes act as a switch that causes the oligomers to form. Stopping the sensor domains from forming oligomers inactivated the IRE1 protein in mammalian cells; this rendered IRE1 unresponsive to stress within the endoplasmic reticulum.

The regulation of IRE1 affects many health disorders, including diabetes, cancer and neurodegenerative diseases. By showing that unfolded proteins switch IRE1 into its active, oligomeric state, these findings might lead to new approaches to manipulate IRE1's activity with small molecules to help to treat these diseases.

DOI: https://doi.org/10.7554/eLife.30700.002

whereas IRE1β expression is restricted to gastrointestinal and respiratory tracts (*Bertolotti et al., 2001*; *Tsuru et al., 2013*). Both IRE1 orthologs are trans-membrane kinase/nucleases that oligomerize in the ER-membrane in response to ER stress (*Aragón et al., 2009*; *Li et al., 2010*). Oligomerization is crucial for IRE1 activation as it allows for *trans*-autophosphorylation and allosteric activation of its endonuclease domain, which for IRE1α then initiates the unconventional splicing of the *XBP1* mRNA (*Aragón et al., 2009*; *Cox et al., 1993*; *Cox and Walter, 1996*; *Korennykh et al., 2009*; *Li et al., 2010*; *Sidrauski and Walter, 1997*; *Yoshida et al., 1998*; *Yoshida et al., 2001*). Spliced *XBP1* mRNA encodes the transcription factor XBP1s, which activates the transcription of several target genes involved in restoring ER homeostasis (*Acosta-Alvear et al., 2007*; *Lee et al., 2003*). While the *XBP1* mRNA is the only known splicing target of IRE1, active IRE1 can also cleave ER-localized mRNAs in a process known as regulated IRE1-dependent decay of messenger RNAs (RIDD), which serves to limit the amount of client proteins entering the ER, thus helping alleviate the folding stress (*Hollien et al., 2009*; *Hollien and Weissman, 2006*).

Two alternative models are used to describe how IRE1's lumenal domain senses ER stress: a recent model where unfolded proteins act directly as activating ligands and an earlier model where IRE1 lumenal domain is indirectly activated through dissociation of the ER-chaperone BiP.

The direct activation model emerged from the crystal structure of the core lumenal domain (cLD) from *S. cerevisiae* IRE1 (yIRE1; 'y' for yeast), where yIRE1 cLD dimers join via a 2-fold symmetric interface IF1$^L$ ('L' for lumenal). A putative peptide-binding groove that architecturally resembles that of the major histocompatibility complexes (MHCs) extends across this interface (*Credle et al., 2005*). yIRE1 selectively binds a misfolded mutant of carboxypeptidase Y (Gly255Arg, CPY*) in vivo, and purified yIRE1 cLD directly interacts with peptides in vitro, leading to its oligomerization. Taken together, these observations support the model that direct binding of unfolded proteins in the ER lumen to IRE1 induces its oligomerization leading to IRE1 activation (*Gardner and Walter, 2011*).

Due to structural differences between human and yeast IRE1 lumenal domains, it is not yet clear if this mechanism is also used by mammalian IRE1. Although the crystal structure of human IRE1α (hIRE1α) cLD displays conserved structural elements in its core, there are several notable differences between the crystal structures of human and yeast IRE1 cLD known to date (*Figure 1*). First, the helices flanking the groove in yIRE1 cLD are too closely juxtaposed in the human structure to allow formation of the MHC-like groove present in the yeast (*Zhou et al., 2006*). Second, the yIRE1 cLD structure displays a second interface, IF2$^L$, which provides contacts for higher order oligomerization, which was experimentally validated to be indispensable for yIRE1 activation in vivo (*Figure 1*). In the yIRE1 cLD, an α-helix–turn region forms an important element in IF2$^L$ making contacts with the incomplete β-propeller in the neighboring protomer. Notably, the residues corresponding to the α-helix–turn are not resolved in the hIRE1α cLD crystal structure (aa V307-Y358). Instead, hIRE1α cLD has two other symmetry mates in addition to the dimerization interface, which appear to be crystal lattice contacts that are predicted to be too energetically unstable to form biologically important oligomerization interfaces (*Zhou et al., 2006*). Indeed, the equivalent of an IF2$^L$ cannot form in the depicted hIRE1α cLD structure because of a steric hindrance from a prominent α-helix ('αB helix'; aa V245-I263) that is absent in yIRE1 cLD (*Figure 1*) (*Zhou et al., 2006*).

The structural differences between IRE1 orthologs were cast to support the indirect model of IRE1 activation in higher eukaryotes (*Zhou et al., 2006*). This model poses that due to the aforementioned structural differences—rather than direct unfolded protein binding—it is the reversible dissociation of the ER-resident Hsp70-type chaperone BiP from IRE1's lumenal domain the main driving force regulating hIRE1α activity (*Zhou et al., 2006*). According to this view, titration of BiP to unfolded proteins upon ER stress licenses IRE1 activation (*Bertolotti et al., 2000*; *Carrara et al., 2015*; *Oikawa et al., 2009*; *Zhou et al., 2006*). In yeast, however, this view has been experimentally refuted (*Kimata et al., 2004*; *Pincus et al., 2010*).

Considering the degree of conservation at various features of IRE1 mechanism of action from yeast to mammals, we favor the unifying direct activation model. Such model finds support in the notion that all structures adopted by a protein in a crystal lattice represent a singular snapshot of many possible conformational states. Therefore, it is entirely plausible that human and yeast IRE1 cLD use a common mechanism of activation and that the divergent structures aforementioned represent different states in a spectrum of possible conformational states that the IRE1 cLD from *any species* could assume. In this scenario, the crystal structure of hIRE1α cLD represents a 'closed' conformation that can shift towards an 'open' state to allow peptide binding in the MHC-like groove that is apparent in the structure of the yeast ortholog (*Video 1*) (*Gardner et al., 2013*; *Gardner and*

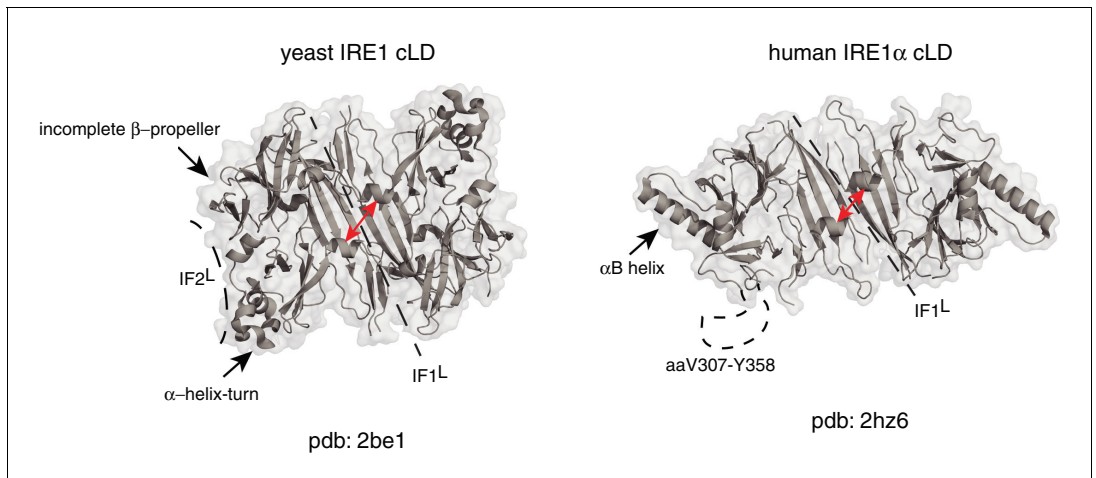

**Figure 1.** Human and yeast IRE1 cLD's crystal structures display distinct features. The αB helix in hIRE1α cLD structure (pdb: 2hz6) , the helix-turn region and the incomplete β-propeller in yIRE1 cLD (pdb: 2be1) structure are indicated with arrows. The interfaces IF1$^L$ and IF2$^L$ and the unresolved dynamic region (aaV307-Y358) in hIRE1α cLD crystal structure are depicted with dashed lines. The distance between the helices surrounding the groove in yIre1 and hIRE1α cLD is depicted with red arrows.
DOI: https://doi.org/10.7554/eLife.30700.003

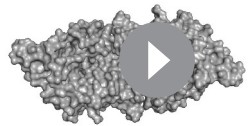

**Video 1.** The model displaying the transition of hIRE1α cLD from the 'closed' to 'open' state. The crystal structure of hIRE1α cLD is used to represent (pdb: 2hz6) the closed and the hIRE1α cLD structural model based on the yeast crystal structure (pdb: 2be1) represents the open state. The movie is generated using Pymol.
DOI: https://doi.org/10.7554/eLife.30700.004

Walter, 2011). As such, this model predicts specific outcomes that can be experimentally tested. Specifically, that (i) human IRE1 α cLD can bind to unfolded polypeptides, (ii) unfolded polypeptide binding stabilizes the open conformation of the hIRE1α cLD, and (iii) the open conformation of hIRE1α cLD favors its oligomerization.

Here, we used complementary biochemical and structural approaches to experimentally explore the mechanism of human IRE1α activation. We show that hIRE1α cLD—just like its yeast ortholog—directly binds select peptides with a characteristic amino acid bias. State-of-the-art NMR experiments that probe dynamic conformational states further support an activation mechanism involving peptide binding to the MHC-like groove and stabilizing the open conformation of hIRE1α cLD. Moreover, we provide insights into the mechanism that couples peptide binding and oligomerization to produce active IRE1 oligomers. Importantly, we show by mutational analysis that lumenal domain driven oligomerization is crucial for IRE1 function in mammalian cells. Taken together, our results resolve the discrepancies between existing models of IRE1 activation and supports a model in which unfolded polypeptides can bind and directly activate human IRE1.

## Results

### The lumenal domain of human IRE1α binds peptides

To test whether, akin to yeast IRE1, mammalian IRE1 also binds unfolded proteins directly, we employed peptide tiling arrays. To identify hIRE1α cLD-binding peptides, we designed tiling arrays utilizing ER-targeted model proteins known to induce the UPR either by overproduction (proinsulin and 8ab protein from SARS-corona virus [*Scheuner et al., 2001*; *Sung et al., 2009*]) or through destabilizing point mutations (myelin protein zero (MPZ)). The peptide arrays were composed by tiling 18-mer peptides that step through the entire protein sequence, shifting by three amino acids between adjacent spots. We incubated the peptide arrays with purified hIRE1α cLD fused N-terminally to maltose-binding protein (MBP) and probed with an anti-MBP antibody. As shown in *Figure 2A* (left panel), MBP-hIRE1α cLD bound a select subset of peptides on the arrays. To maximize the available sequence space, we analyzed binding of MBP-hIRE1α cLD to these peptides irrespective of their topological accessibility in the ER lumen. hIRE1α cLD recognized peptide sequences found in both the ER-lumenal and cytosolic domains of MPZ, which we considered together in our analyses to define the chemical properties of cLD peptide recognition. We found that hIRE1α cLD-binding peptides with the top 10% binding scores were enriched in cysteine, tyrosine, tryptophan, and arginine (*Figure 2B*, *Figure 2—figure supplement 1A*, p<0.05). By contrast, aspartate and glutamate were strongly disfavored, together with glutamine, valine, and serine.

### IRE1 and the ER-resident chaperone BiP recognize a different subset of peptides

At a first glance, the amino acid preferences displayed by mammalian IRE1 cLD resemble those of the other chaperones including the ER chaperone BiP (*Blond-Elguindi et al., 1993*; *Deuerling et al., 2003*; *Flynn et al., 1991*). Like BiP, hIRE1α cLD favored binding to aromatic and positively charged residues (*Blond-Elguindi et al., 1993*; *Otero et al., 2010*). BiP is a highly abundant chaperone in the ER lumen, whereas IRE1 is present at orders of magnitude lower levels (*Ghaemmaghami et al., 2003*). Therefore, if IRE1 and BiP recognize the same regions of unfolded proteins, the peptide-binding activity of hIRE1α cLD would depend entirely on saturation of BiP by unfolded substrate proteins—a scenario difficult to reconcile with IRE1's task of dynamically sensing ER stress. To address

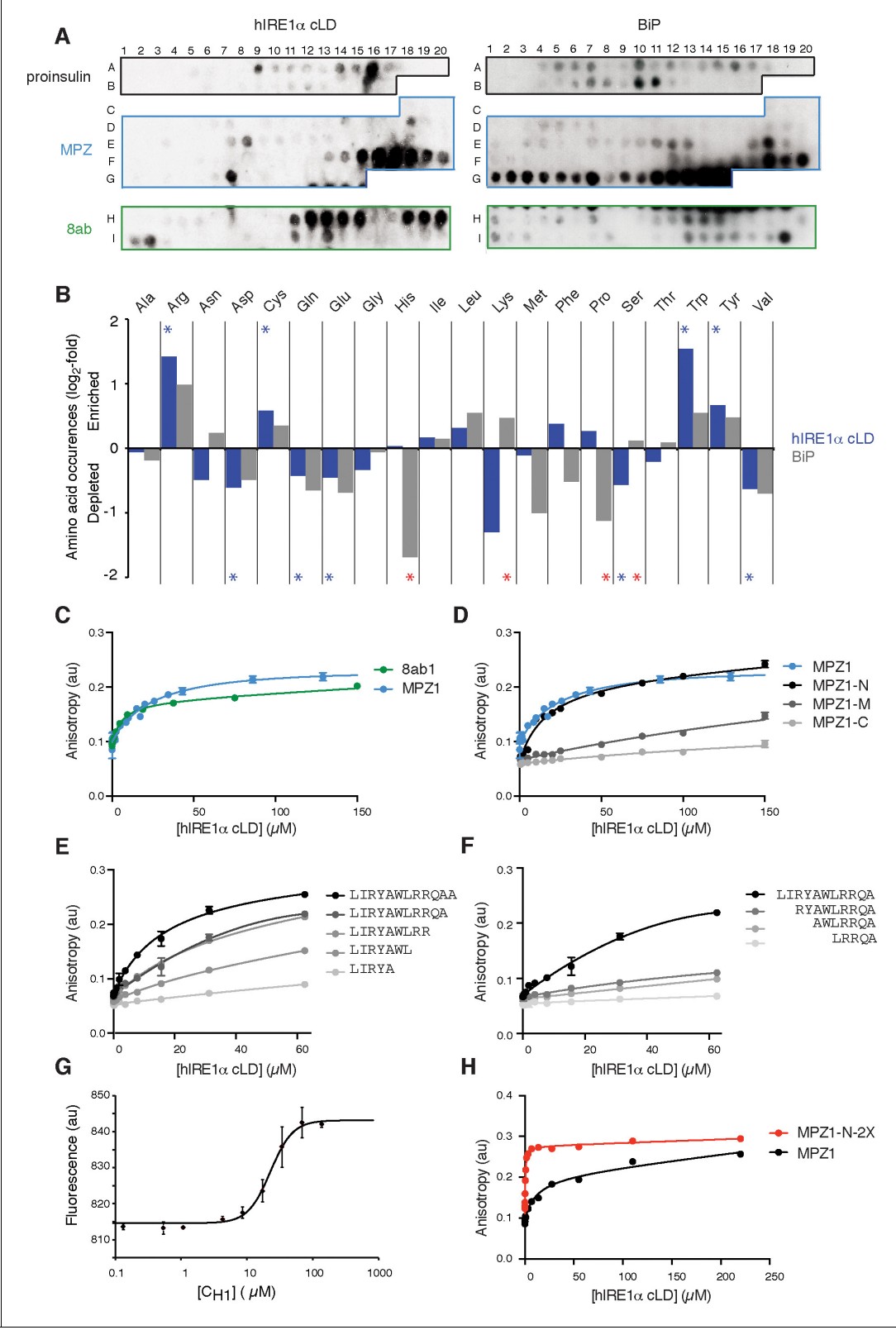

**Figure 2.** hIRE1α cLD binds peptides and unfolded proteins  (A) Peptide arrays tiled with 18mer peptides derived from proinsulin, myelin protein zero (MPZ), 8ab are probed with MBP-hIRE1α cLD (on the left) or His$_{10}$-BiP (on the right). (B) Comparison of the amino acid preferences of MBP-hIRE1α cLD (blue) and His$_{10}$-BiP (gray). The peptide arrays were quantified using Max Quant. The binding intensity in each spot was normalized to max signal intensity in the peptide array. The peptides with the top 10% binding scores were selected and the occurrence of each amino acid in these top-binding

*Figure 2 continued on next page*

*Figure 2 continued*

peptides was normalized to their total abundance in the arrays. The normalized occurrences are plotted in $\log_2$ scale. Blue stars depict the amino acids that are significantly enriched or depleted in hIRE1α cLD binders ($p<0.05$), whereas red stars depict differences in binding preferences of hIRE1α cLD and BiP ($p<0.05$). (C) hIRE1α cLD binds peptides derived from proteins MPZ, 'MPZ1' (in blue) (peptides F16-F17 in *Figure 1a*, sequence: LIRYCWLRRQAALQRRISAME) and 8ab, '8ab1' (in green) (peptide H20 in *Figure 1a*, sequence: WLCALGKVLPFHRWHTMV with a $K_{1/2}$ of 24 ± 4.7 μM and 5 ± 1.7 μM, respectively, determined by fluorescence anisotropy measurements. (D) Fluorescence anisotropy measurements show that N-terminal 12mer derivative of MPZ1 peptide, 'MPZ1-N' binds to hIRE1α cLD with a similar affinity as the full-length peptide, with a $K_{1/2=}$ 16.0 ± 2.6 μM. The binding curves of N-terminal (MPZ1-N, sequence: LIRYCWLRRQAA), Middle (MPZ1-M, sequence: WLRRQAALQRR) and C-terminal (MPZ1-C, sequence: LQRRISAME) fragments are shown in black, dark gray, and light gray, respectively. (E) The binding affinity of C-terminal truncations of MPZ1-N for hIRE1α cLD was measured by fluorescence anisotropy. The binding curves for the truncated peptides are shown in different shades of gray. (F) Fluorescence anisotropy measurements with N-terminal truncations of the MPZ1-N peptide are shown in different shades of gray. (G) IRE1 cLD binds to unfolded $C_H1$ domain of IgG1 with a $K_{1/2}$ of 29.2 ± 1.2 μM determined by microscale thermophoresis measurements. (H) Fluorescence anisotropy measurements show that MPZ1-N-2X peptide where MPZ1-N peptide sequence is repeated twice in the peptide binds tighter to hIRE1α cLD (peptide sequence: LIRYAWLRRQAALQRRLIRYAWLRRQAA).

DOI: https://doi.org/10.7554/eLife.30700.005

The following figure supplements are available for figure 2:

**Figure supplement 1.** hIRE1α cLD shows preference for arginines and aromatic residues.
DOI: https://doi.org/10.7554/eLife.30700.006

**Figure supplement 2.** hIRE1α cLD binds unfolded proteins.
DOI: https://doi.org/10.7554/eLife.30700.007

this point, we compared the binding preferences of mammalian BiP (fused to an N-terminal 10x-histidine tag) on the same peptide arrays. We found sequences recognized by both hIRE1α cLD and BiP (*Figure 2A*, *Figure 2—figure supplement 1A,B*). Importantly, however, we also found profound differences. While IRE1 tolerated both prolines and histidines, BiP strongly disfavored these amino acids (*Figure 2B*, *Figure 2—figure supplement 1A*, $p<0.05$). Conversely, BiP tolerated serine and threonines, while IRE1 strongly disfavored them. Thus, IRE1 can recognize regions of unfolded proteins to which BiP would not readily bind and *vice versa*, thereby providing a plausible explanation of how IRE1 could recognize unfolded proteins despite of the vast excess of BiP over hIRE1α LD in the ER.

## hIRE1α cLD binds peptides with distinct biochemical properties

To measure binding affinities of hIRE1α cLD's interaction with peptides in solution, we selected the two peptides with the highest binding scores in the peptide arrays (MPZ- and 8ab-derived peptides, henceforth referred to as 'MPZ1' and '8ab1', respectively) and attached fluorophores at their N-termini. Fluorescence anisotropy revealed that hIRE1α cLD bound to MPZ1 with $K_{1/2}$ = 24 ± 4.7 μM and to 8ab1 with $K_{1/2}$ = 5 ± 1.7 μM (*Figure 2C*). (Note that we used $K_{1/2}$ to denote a measure of affinity because, as we show below, hIRE1α cLD exists in solution as an ensemble of different interconverting conformational states and our measurements therefore score several superimposed equilibria. The measured affinities therefore do not reflect true $K_d$ values). These affinities fall within the same order of magnitude of chaperone binding to unfolded proteins, supporting the notion that similar modes of fast transient interactions with unfolded proteins are adopted by both IRE1 and chaperones (*Karagöz et al., 2014*; *Marcinowski et al., 2011*; *Street et al., 2011*).

To identify the minimal region in MPZ1 for binding to hIRE1α cLD, we next divided MPZ1 into 12, 11 and 9 amino acid long fragments representing its N-terminal (MPZ1-N), middle (MPZ1-M) and C-terminal (MPZ1-C) regions and measured their respective affinities for hIRE1α cLD. hIRE1α cLD bound to MPZ1-N with a similar affinity as the full-length peptide ($K_{1/2}$ = 16.0 ± 2.6 μM, *Figure 2D*), whereas the other peptide fragments displayed much lower binding affinities ($K_{1/2}$ = 377 ± 54 μM and 572 ± 107 μM, respectively, assuming similar maximum anisotropy values as for the MPZ1-N peptide). We further truncated MPZ1-N by two residues at a time from either its N- or C-terminus. Deleting amino acids from the C-terminus gradually decreased the affinity (*Figure 2E*). By contrast, deletion of the first two hydrophobic residues from the N-terminus (leucine and isoleucine) abolished its binding to hIRE1α cLD (*Figure 2F*). These analyses revealed that the minimum peptide length with a comparable binding affinity to the full-length MPZ1 peptide is a 12-mer. This 12-mer peptide matches the chemical properties we found for hIRE1α cLD-binding peptides: it is enriched in

aromatics, hydrophobic amino acids and arginines, indicating that specific binding contacts play a role in hIRE1α cLD's interaction with unfolded proteins.

## hIRE1α cLD binds unfolded proteins

To validate that peptides are valid surrogates for unfolded proteins, we next tested binding of intact but constitutively unfolded proteins to hIRE1α cLD. Immunoglobulins (IgGs) mature in the ER using a well-characterized folding pathway, wherein the constant region domain of the IgG heavy chain ($C_{H1}$) remains disordered until it binds to its cognate partner, the constant region domain of the IgG light chain $C_L$ (*Feige et al., 2009*). We measured the binding affinity of $C_{H1}$ to hIRE1α cLD by thermophoresis, which reports on changes in the hydration shell of a biomolecule upon interaction with a partner in solution (*Jerabek-Willemsen et al., 2011*). By contrast to earlier findings that showed no measurable binding of hIRE1α cLD to $C_{H1}$ under different experimental conditions (*Carrara et al., 2015*), our experiments showed that hIRE1α cLD interacts with $C_{H1}$ with a $K_{1/2}$ = 29.2 ± 1.2 μM (*Figure 2G*). To further validate this observation, we measured binding of hIRE1α cLD to another model unfolded protein by fluorescence anisotropy, the folding mutant of staphylococcal nuclease Δ131Δ (*Street et al., 2011*). We observed a comparable binding affinity of $K_{1/2}$ = 21.4 ± 2.3 μM (*Figure 2—figure supplement 2*). Our data thus show that hIRE1α cLD binds to full-length unfolded proteins with similar affinity as peptides, suggesting that these proteins display a distinct single binding site for hIRE1α cLD.

To test whether multiple binding sites would increase the affinity for hIRE1α cLD, we synthesized a peptide consisting of two MPZ1-N tandem repeats separated by a 5-amino acid spacer (MPZ1-N-2X). Intriguingly, MPZ1-N-2X bound to hIRE1α cLD with an order of magnitude higher affinity ($K_{1/2}$ = 0.456 ± 0.07 μM) compared to MPZ1 peptide (*Figure 2H*). As we show below, the increased apparent affinity is due to avidity of hIRE1α cLD to the peptide.

## hIRE1α cLD is structurally dynamic

To capture evidence for structural rearrangements in hIRE1α cLD predicted by a switch-mechanism that oscillates between inactive closed and active open conformations as we suggest in the Introduction, we employed nuclear magnetic resonance (NMR) spectroscopy. NMR spectroscopy reveals structural information at the atomic level for dynamic protein complexes and is well suited to study structural changes in hIRE1α cLD upon its interaction with peptides and unfolded proteins. The hIRE1α cLD dimer is ~80 kDa and thus is well above the size limit for conventional NMR approaches. We therefore used methyl transverse relaxation optimized spectroscopy (methyl-TROSY), a specific NMR method that allows to extract structural information from large proteins after selective isotopic labeling of side chain methyl groups with carbon-13 ($^{13}$C) (*Tugarinov et al., 2004*; *Tugarinov et al., 2007*) in select amino acids including isoleucines. hIRE1α cLD has 12 isoleucines per monomer, which are evenly distributed throughout the protein (*Figure 3A*). In hIRE1α cLD's methyl-TROSY spectra, we resolved seven peaks corresponding to isoleucines (*Figure 3B*), which then served as sensors of peptide binding and accompanying conformational changes. All isoleucine peaks in hIRE1α cLD's NMR spectrum displayed broad line widths (*Figure 3B*), which is indicative of chemical exchange resulting from hIRE1α cLD sampling multiple conformational states at the conditions of the NMR experiments. These data revealed that hIRE1α cLD is dynamic in solution.

To assign the resolved peaks to specific amino acids in the hIRE1α cLD sequence, we mutated each isoleucine to leucine, alanine or valine and monitored the disappearance of each resolved peak in methyl-TROSY spectra of the mutant proteins. This approach allowed us to assign six isoleucine peaks unambiguously (*Figure 3C,D*, *Figure 3—figure supplements 1*, *2* and *3*). To further increase the number of NMR visible probes in hIRE1α cLD, we mutated Leu186 and Thr159 to isoleucines (*Figure 3E,F*, *Figure 3—figure supplement 1C and F*). Leu186 lies in an amphipathic unstructured loop surrounding the putative groove in hIRE1α cLD. The Leu186Ile peak displayed high signal intensity consistent with a dynamic and flexible position (*Figure 3F*, *Figure 3—figure supplement 1F*). By contrast, Thr159 lies at the β-sheet floor in hIRE1α cLD structure where its side chain faces towards the MHC-like groove and, as expected, the Thr159Ile substitution resulted in a low-intensity peak (*Figure 3—figure supplement 1C*).

We further enhanced the coverage of hIRE1α cLD with NMR-visible probes in complementary experiments in which we labeled threonine side chains with $^{13}$C at their γ$_2$ methyl groups (*Figure 3—*

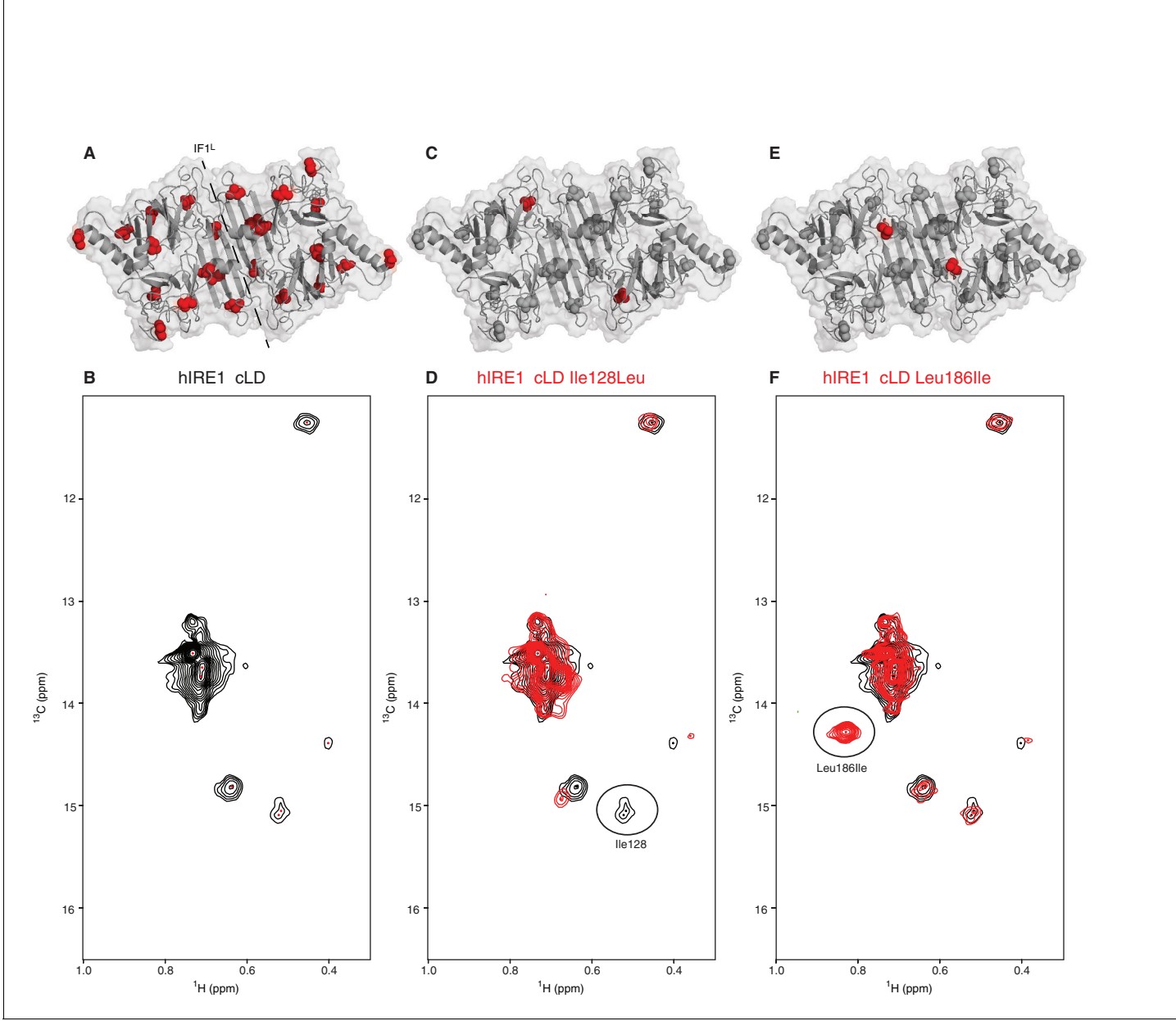

**Figure 3.** NMR spectroscopy reveals dynamic nature of hIRE1α cLD (A) Isoleucines serving as probes in the NMR experiments are evenly distributed throughout hIRE1α cLD. hIRE1α cLD structural model is shown in gray, with space-filling isoleucine side chains shown in red. The structural model of hIRE1α cLD was generated by I-Tasser webserver using hIRE1α cLD crystal structure (pdb:2hz6) as a template to visualize the loops that are not resolved in the crystal structure (*Roy et al., 2010*; *Zhang, 2008*). The dimerization interface IF1$^L$ of hIRE1α cLD is depicted with a dashed line. (B) Methyl-TROSY spectrum of hIRE1α cLD with selective $^{13}$C labeling at $\delta_1$- methyl group of isoleucines resolves seven peaks, indicated by red dots. (C) Ile128 is highlighted as red spheres on hIRE1α cLD structural model. hIRE1α cLD is shown in gray, with isoleucine side chains are depicted as grayspace-fillings. (D) Assignment strategy for isoleucines in hIRE1α cLD. The WT hIRE1α cLD spectrum (in black) is overlaid with the spectrum of Ile128Ala mutant (depicted in red). The signal that disappeared in the mutant spectrum corresponds to Ile128 peak and is depicted with a circle. (E) The side chain of Leu186 is highlighted as red spheres on hIRE1 α cLD structural model. (F) The WT hIRE1α cLD spectrum (in black) is overlaid with the spectrum of Leu186Ile mutant (depicted in red). The signal that appeared in the mutant spectrum that corresponds Leu186Ile peak is depicted with a circle.

DOI: https://doi.org/10.7554/eLife.30700.008

The following figure supplements are available for figure 3:

**Figure supplement 1.** Assignment strategy of the isoleucines in hIRE1α cLD spectrum.
DOI: https://doi.org/10.7554/eLife.30700.009
**Figure supplement 2.** Assignment strategy of the isoleucines in hIRE1α cLD spectrum.

*Figure 3 continued on next page*

*Figure 3 continued*

DOI: https://doi.org/10.7554/eLife.30700.010

**Figure supplement 3.** The assigned isoleucines in wild type hIRE1α cLD are depicted on the spectrum.

DOI: https://doi.org/10.7554/eLife.30700.011

**Figure supplement 4.** Methyl-TROSY spectrum of hIRE1α cLD with selective $^{13}C$ labeling at $\gamma_2$ methyl group of threonines resolves 24 residues.

DOI: https://doi.org/10.7554/eLife.30700.012

*figure supplement 4*). There are 33 threonine residues in hIRE1α cLD, 24 of which were detected by the NMR experiments. While we did not assign threonine peaks in hIRE1α cLD spectrum due to high spectral crowding, they provided an additional 'fingerprint' reporting on peptide binding-induced changes in hIRE1α cLD.

## Peptide binding stabilizes the open conformation of hIRE1α cLD

Next, we used methyl-TROSY experiment to monitor changes in the environment of isoleucines and threonines in hIRE1α cLD upon peptide binding. A largely overlapping subset of isoleucine and threonine peaks shifted when hIRE1α cLD bound to the peptides MPZ1 or 8ab1, indicating a change in a localized environment upon peptide binding (*Figure 4A,B*, *Figure 4—figure supplement 1A–C*). Yet, a subset of isoleucine and threonine peaks displayed peptide specific changes. The chemical shifts displayed by the isoleucine peaks were not very large yet reproducible upon binding of different peptides allowing us to probe peptide induced changes in hIRE1α cLD. By contrast, the threonine peaks displayed larger chemical shifts, which is expected from their higher solvent exposure rendering them more sensitive to binding events (*Figure 4B*, *Figure 4—figure supplement 1B and C*). Mapping the chemical shift perturbations of the isoleucine peaks upon peptide binding on the hIRE1α cLD structure (*Figure 4A–E*, *Figure 4—figure supplement 1B*, *Figure 4—figure supplement 2*) revealed that the isoleucine resonances that shifted most significantly lie on the floor of the central β-sheet (marked by Ile124, Ile128, Thr159Ile) (*Figure 4D and E*). Among these isoleucines, only the side chain of Thr159Ile faces towards the MHC-like groove. We noted that in comparison to the isoleucines 124 and 128, Thr159Ile peak displayed a larger shift upon peptide binding (*Figure 4C and E*, *Figure 4—figure supplement 1D*). In addition to the central β-sheet floor, the αB helix that lies at the ends of hIRE1α cLD dimer (marked by Ile263), and the β-sandwich connecting the β-sheet floor to the αB helix (marked by Ile52) were affected, albeit to a lesser extent. By contrast, the unstructured loop extending from the MHC-like groove (marked by Ile186) was only slightly affected and the isoleucines positioned in the flexible region that are not resolved in the crystal structure (marked by Ile326 and Ile334) did not shift (*Figure 4D,E*).

Importantly, binding of the unfolded protein $C_{H1}$ shifted the same peaks in the hIRE1α cLD spectra as the short peptides suggesting peptides and unfolded protein chains interact with hIRE1α cLD in a similar way (*Figure 4F*, *Figure 4—figure supplement 2*). Taken together, these results indicate that peptide as well as unfolded protein binding populate a distinct conformational state of hRE1α cLD, consistent with a peptide-induced closed-to-open conformational transition. Moreover, the results are consistent with a model in which peptide binding induces conformational changes that propagate from the MHC-like groove via the β-sandwich to affect the regions involved in oligomerization.

## Peptide binding maps to the MHC-like groove in hIRE1α cLD

To map the peptide-binding site in hIRE1α cLD with higher precision, we employed paramagnetic relaxation enhancement (PRE) experiments (*Gaponenko et al., 2000*; *Gillespie and Shortle, 1997*) using MPZ1 modified with a nitroxide spin label, 3-(2-Iodoacetamido)-PROXYL, at cysteine residue, Cys5 (*Figure 5A,B*). The unpaired electron in the spin label broadens (in a range of 1 to 2.5 nm) or entirely erases (distances <1 nm) NMR signals in its vicinity in a distance dependent manner (*Gottstein et al., 2012*). Binding of the spin label attached peptide to hIRE1α cLD would result in a decrease in the intensity of isoleucine peaks depending on their relative distance to the peptide-binding site. Therefore, we analyzed the changes in the intensities of all isoleucine signals upon binding of MPZ1-proxyl peptide to hIRE1α cLD (*Figure 5B*, *Figure 5—figure supplement 1A–C*). Binding of MPZ1-proxyl to hIRE1α cLD erased the otherwise very strong signal of Leu186Ile and

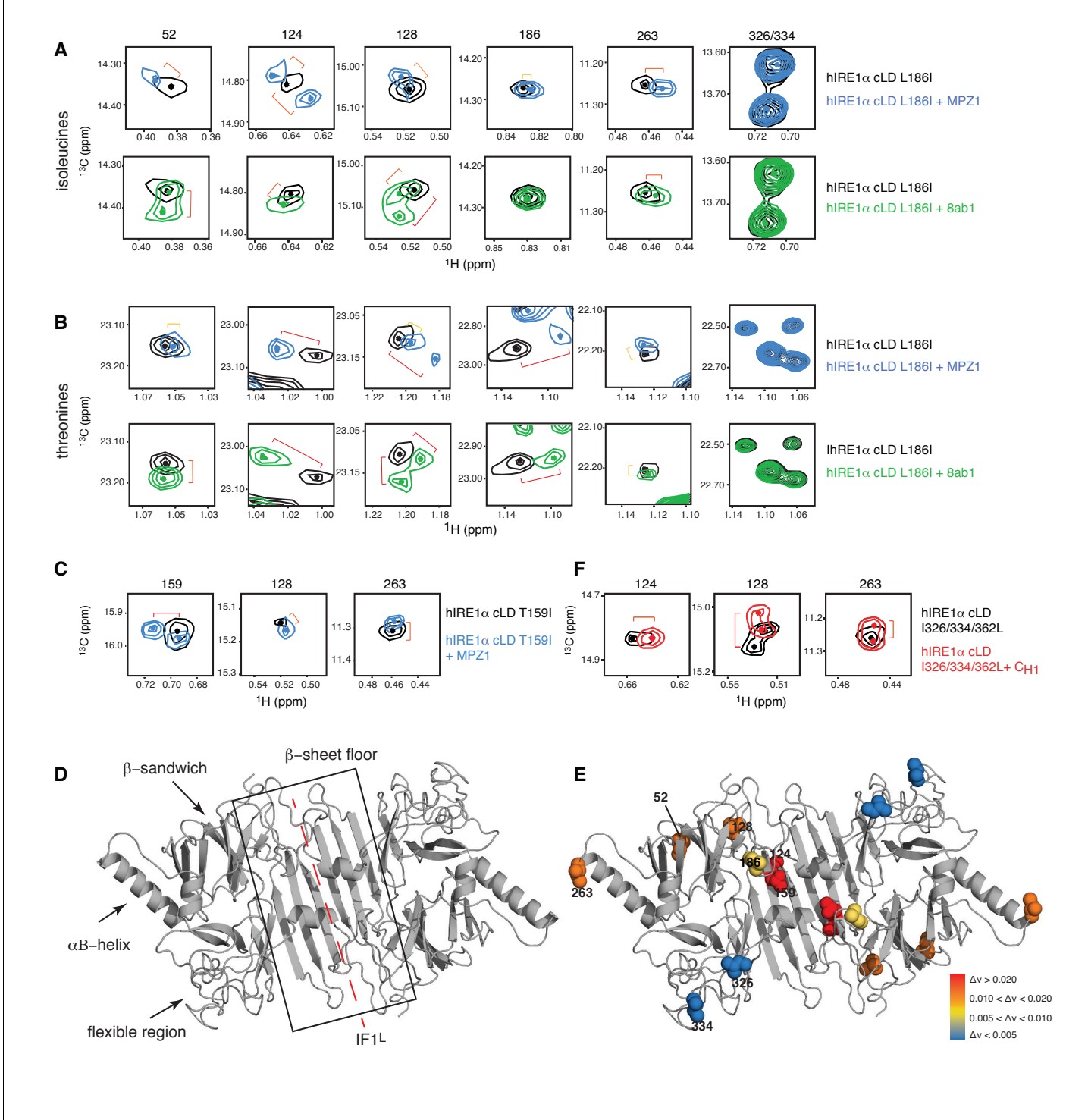

**Figure 4.** Peptide binding induces conformational changes in hIRE1α cLD (A) Close-up of the isoleucine peaks in the methyl-TROSY spectrum of hIRE1α cLD Leu186Ile alone (black, 50 μM) and of hIRE1α cLD Leu186Ile bound to MPZ1 (1:1 molar ratio) (upper panel, blue), or to 8ab1 peptide (lower panel, green, 1:1 molar ratio) shows that peptide binding shifts select peaks. The shift of each peak is indicated with brackets consistent with the color code in Figure 4—figure supplement 1B, where yellow color indicates chemical shift perturbation values Δν > 0.005, orange, Δν > 0.010 and red Δν > 0.020. The identities of isoleucine peaks are indicated on top of each peak. (B) Close-up of the threonine peaks in the methyl-TROSY spectrum of hIRE1α cLD Leu186Ile alone (black, 50 μM) and of hIRE1α cLD Leu186Ile bound to MPZ1 (1:1 molar ratio) (upper panel, blue), or to 8ab1 peptide (lower panel, green, 1:1 molar ratio) shows that peptide binding shifts select peaks upon binding of peptides. The chemical shift of each peak is indicated as in *Figure 4A* based on the chemical shift perturbations calculated in *Figure 4—figure supplement 1C*. (C) Close-up of select isoleucine peaks in the

*Figure 4 continued on next page*

Figure 4 continued

methyl-TROSY spectrum of hIRE1α cLD T159I mutant alone (black, 25 μM) and in the spectrum of hIRE1α cLD bound to MPZ1-N peptide (1:1 molar ratio) (blue). The chemical shift of each peak is indicated with brackets consistent with the color code in *Figure 4—figure supplement 1B,D*. (D) Important structural regions of hIRE1 cLD are depicted on the structural model of hIRE1α cLD by arrows. The red dashed-lines indicate the dimerization interface IF1$^L$ of hIRE1α cLD, whereas the black box shows the β-sheet floor of the MHC-like groove. (E) The isoleucine peaks shifting upon MPZ1 binding are mapped into the hIRE1α cLD structural model based on their combined chemical shift perturbation values as shown in Figure 4—figure supplements 4 and 6. The red spheres indicate isoleucine peaks with significant shifts (Δν > 0.020), orange; moderate shifts Δν > 0.010, and yellow spheres show isoleucines that shift slightly upon peptide binding, Δν > 0.005. The isoleucine peaks that do not change significantly (Δν < 0.005) are depicted in blue. (F) Close-up of select isoleucine peaks in the methyl-TROSY spectrum of hIRE1α cLD triple mutant Ile326/334/362Leu alone (black, 50 μM) overlaid with the spectrum of hIRE1α cLD when bound to C$_{H1}$ domain (1:1 molar ratio) (red). The shifts are indicated with brackets consistent with color coding in *Figure 4B* and *Figure 4—figure supplement 1B,D*.

DOI: https://doi.org/10.7554/eLife.30700.013

The following figure supplements are available for figure 4:

**Figure supplement 1.** Peptide induced conformational changes in hIRE1α cLD.

DOI: https://doi.org/10.7554/eLife.30700.014

**Figure supplement 2.** Chemical shift perturbation analysis of isoleucine signals upon binding of C$_{H1}$ domain to hIRE1α cLD I326-334-362L triple mutant (based on the spectrum in *Figure 4f*).

DOI: https://doi.org/10.7554/eLife.30700.015

broadened that of Ile124 (*Figure 5B–D* and *Figure 5—figure supplement 1A–C*). Importantly, Ile128 and Ile263 signals, which shifted upon MPZ1 binding as discussed above (*Figure 4E*), broadened to a lesser extent, suggesting that these residues lie further from the peptide-binding site (*Figure 5—figure supplement 1C and D*). Their resonances therefore shifted due to peptide-induced distant conformational rearrangements. Displaying the normalized PRE effect on hIRE1α cLD structure revealed that MPZ1-proxyl binding mapped to the center of the MHC-like groove, suggesting that peptides bind to MHC-like groove and induce distant conformational changes in hIRE1α cLD (*Figure 5D*).

## Peptide binding induces oligomerization of hIRE1α cLD

To test whether the distant conformational changes in hIRE1α cLD monitored by the NMR experiments are due to peptide binding-induced oligomerization, we employed analytical ultracentrifugation (AUC) sedimentation velocity experiments to assess the oligomeric status of hIRE1α cLD in the absence and presence of peptides. At the concentration range used at NMR experiments (25–75 μM), hIRE1α cLD was found as a mixture of various oligomeric states, where the main peaks corresponded to dimers and tetramers (with higher amount of tetramers formed at higher concentrations, see *Figure 6A*). Notably, binding of MPZ1-N to hIRE1α cLD at the NMR concentrations sharpened the tetramer peak and induced formation of larger oligomeric species in these experiments (*Figure 6A*). The peptide concentration used in these experiments does not saturate hIRE1α cLD molecules based on a determined K$_{1/2}$ of 16.0 ± 2.6 μM, therefore only a small population of hIRE1α cLD formed higher oligomers (depicted as the pink area) (*Figure 6A*).

To assess hIRE1α cLD's oligomeric status at varying hIRE1α cLD concentrations, we performed size exclusion chromatography and found that hIRE1α cLD eluted at earlier fractions in a concentration-dependent manner (*Figure 6—figure supplement 1A*). AUC data confirmed these findings and showed that at concentrations close to its dimerization constant of 2.5 μM, hIRE1α cLD sediment as a single peak with a sedimentation coefficient corresponding to a mixture of monomers and dimers (*Figure 6B*, *Figure 6—figure supplement 1B*). In this concentration regime (from 1 to 2.5 μM), the hIRE1α cLD peak progressively shifted to higher sedimentation values with increasing hIRE1α cLD concentration (*Figure 6—figure supplement 1B*). Peptide binding to hIRE1α cLD shifted the hIRE1α cLD population to even higher sedimentation values (*Figure 6B*, blue trace), indicating that under these conditions peptide binding stabilized hIRE1α cLD dimers and lead to the formation of oligomers.

## Oligomerization leads to global conformational changes in hIRE1α cLD

As hIRE1α cLD populated distinct oligomeric states in a concentration-dependent manner, we next compared the conformational state of hIRE1α cLD at 5 μM (no higher-order oligomer formation

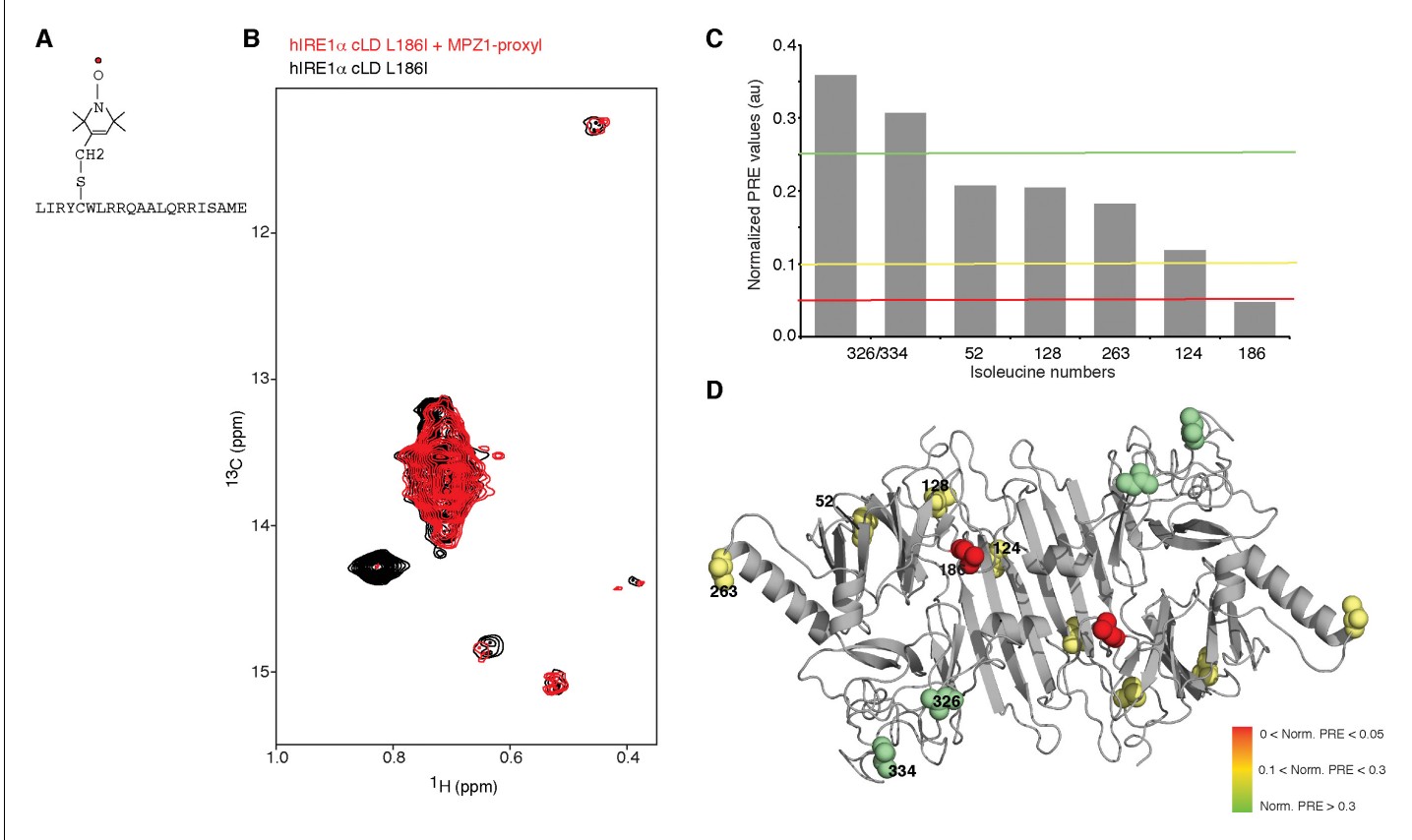

**Figure 5.** Peptide binding maps to the center of MHC-like groove  (A) Schematic representation of the spin label attached MPZ1 peptide. (B) Comparison of the methyl-TROSY spectra of hIRE1α cLD Leu186Ile in the absence (black, 75 μM) and presence of spin-labeled MPZ1 peptide (red, 1:1 molar ratio). (C) The normalized PRE effect on isoleucine peaks upon binding of spin-labeled peptide. The intensity of isoleucine peaks upon MPZ1-proxyl binding is divided by their intensity in the reference spectrum ($I_{PRE}/I_0$) (**Figure 5—figure supplement 1C**) and further normalized to their surface exposed area to exclude possible contributions from non-specific interactions with the spin label attached peptide (**Clore and Iwahara, 2009**) (see Materials and methods). (D) The normalized PRE effect is mapped on the structural model of hIRE1α cLD. The isoleucine peaks in hIRE1α cLD that are broadened upon peptide binding are depicted with a color gradient from red to green as space filling side-chains in the hIRE1α cLD structural model based on decreasing degree of broadening using normalized PRE effect in **Figure 5C**.

DOI: https://doi.org/10.7554/eLife.30700.016

The following figure supplement is available for figure 5:

**Figure supplement 1.** Peptide binding maps to the center of the MHC-like groove in hIRE1α cLD.

DOI: https://doi.org/10.7554/eLife.30700.017

detected by AUC) to 50 μM (based on **Figure 6C**, approximately 60% higher-order oligomer formation) by NMR spectroscopy to probe for the structural differences assumed by these two distinct states (**Figure 6C,D**). In these experiments, we relied on the high sensitivity of selective isoleucine labeling strategy, which could readily detect hIRE1α cLD signals at concentrations as low as 5 μM (**Figure 6—figure supplement 2A,B**).

Notably and similar to effects observed upon peptide binding, oligomerization changed the environment of the αB helix (marked by Ile263) and the β-sandwich connecting the β-sheet floor to the αB helix (marked by Ile52) that both lie at the tips of hIRE1α cLD dimers (**Figure 6D,E**, **Figure 6— figure supplement 2C**). These data suggest that these isoleucines are part of the oligomerization interface and/or that their conformational rearrangements are coupled to the formation of the interface. Moreover, NMR experiments showed chemical shifts in the isoleucines on the beta sheet floor of the groove (marked by Ile124 and Ile128) upon formation of higher oligomers (**Figure 6E**). These coupled, global conformational differences observed by NMR strongly underscore the notion that oligomeric hIRE1α cLD adopts an active conformation and displays higher affinity for unfolded

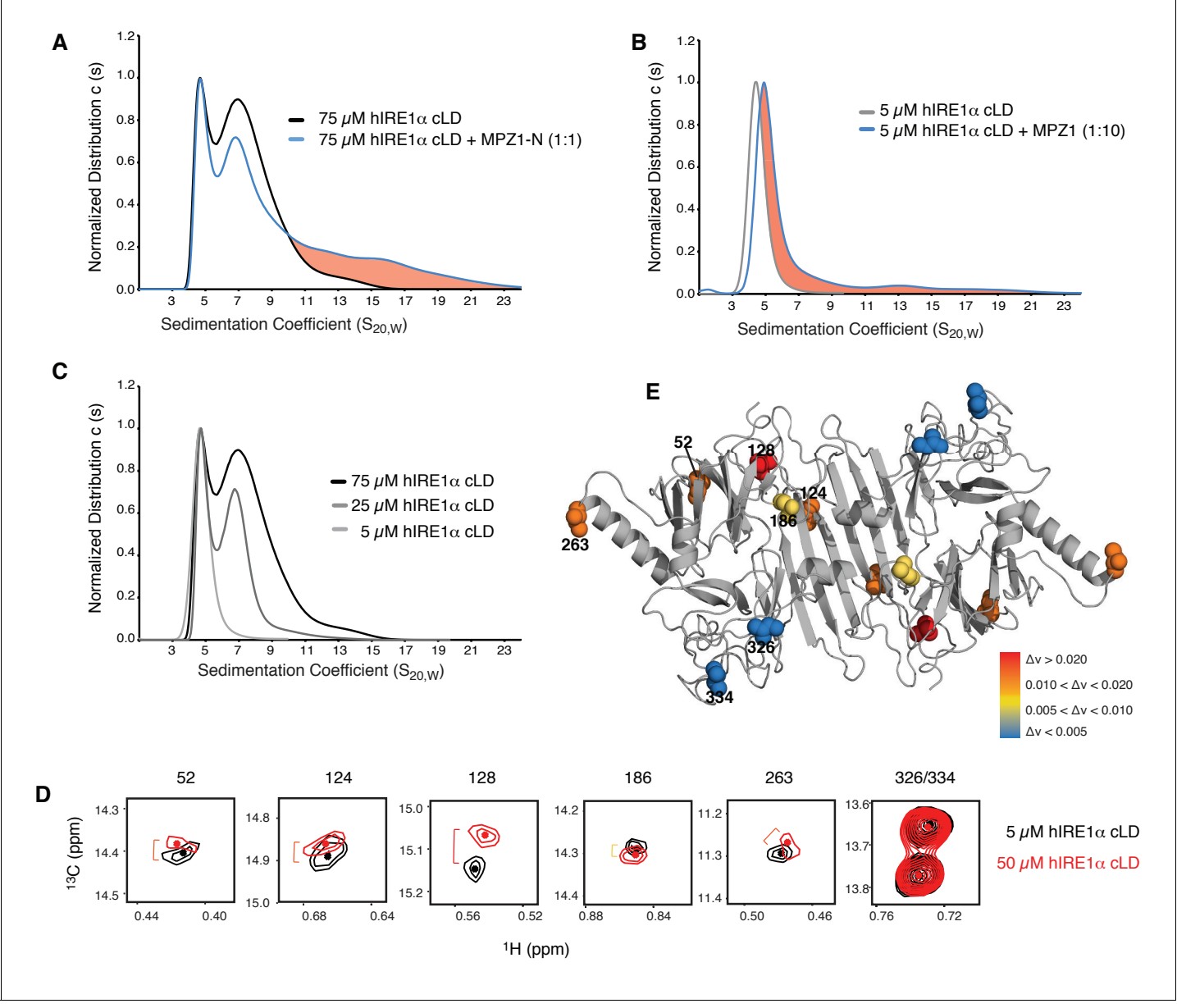

**Figure 6.** hIRE1α cLD forms dynamic oligomers  (A) AUC sedimentation velocity measurements of hIRE1α cLD alone (75 μM) (gray line) versus hIRE1α cLD with 75 μM MPZ1-N peptide (blue line). Pink region indicates larger hIRE1α cLD oligomers formed upon peptide binding. (B) AUC sedimentation velocity measurements of hIRE1α cLD alone (5 μM) (gray line) versus hIRE1α cLD with 50 μM MPZ1 peptide (blue line), pink region indicates the shift in the AUC profile upon peptide binding. (C) AUC sedimentation velocity measurements of hIRE1α cLD at 5, 25 and 75 μM are shown in different shades of gray. (D) Close-up of isoleucine peaks in the methyl-TROSY spectrum of hIRE1α cLD at 5 μM (black) overlaid with the spectrum of hIRE1α cLD at 50 μM (red). (E) The isoleucine peaks shifting upon oligomerization are mapped into the hIRE1α cLD structure based on the chemical shift perturbation values shown in *Figure 6—figure supplement 2C*. The red spheres indicate isoleucine peaks that display most significant shifts (Δν > 0.020), orange; moderate shifts Δν > 0.010, and yellow; slight shifts Δν > 0.005. The isoleucine peaks that do not change significantly (Δν < 0.005) are depicted in blue.

DOI: https://doi.org/10.7554/eLife.30700.018

The following figure supplements are available for figure 6:

**Figure supplement 1.** hIRE1α cLD forms oligomers.

DOI: https://doi.org/10.7554/eLife.30700.019

**Figure supplement 2.** Oligomerization leads to global conformational changes in hIRE1α cLD.

DOI: https://doi.org/10.7554/eLife.30700.020

protein ligands. To address this notion, we set out to experimentally determine the oligomerization interface and then impair the oligomerization of hIRE1α cLD by mutation.

## Identifying the oligomerization interface of hIRE1α cLD

We employed a chemical cross-linking strategy coupled to mass spectrometry to experimentally determine residues that map to the oligomerization interface in hIRE1α cLD. To this end, we cross-linked hIRE1α cLD in the presence and absence of peptides by a homobifunctional cross-linker, BS3 (bis(sulfosuccinimidyl)suberate), which crosslinks primary amines mainly present in lysine side chains. Denaturing SDS-PAGE analysis of hIRE1α cLD after cross-linking revealed that cross-linking captured oligomeric hIRE1α cLD (*Figure 7A*, *Figure 7—figure supplement 1*). We separately isolated the bands corresponding to hIRE1α cLD monomers, dimers and higher oligomers from the gel and analyzed peptides by mass spectrometry. We identified cross-linked peptides by accurate mass measurement of both candidate peptides and their fragment ions (*Chu et al., 2010*; *Trnka et al., 2014*). In comparative analyses, we separated intra- from inter-molecular cross-links by focusing on peaks that were present only in the covalent dimers and higher oligomers (*Wu et al., 2013*; *Zeng-Elmore et al., 2014*). These analyses revealed five abundant cross-links between lysines 120•120, 53•347, 53•349, 53•351 and 265•351 (*Figure 7A*, *Table 1*).

Previous studies of BS3-cross-linked proteins with known crystal structures established that the distance between the αC atoms of cross-linked lysines is less than 28 Å for most cross-links but can be up to 33 Å for a few cases due to local protein flexibility (*Leitner et al., 2010*), in agreement with the additive lengths of the cross-linker itself plus twice the length of the lysine side chain. The Lys120•120 cross-link maps to hIRE1α cLD's dimerization interface (IF1[L]), whereas the four other cross-links are compatible with being positioned at hIRE1α cLD oligomerization interface, IF2[L]. The cross-links Lys53•347, Lys53•349, Lys53•351 and Lys265•351 each involve one lysine residue (Lys53 and Lys263) that is close to the isoleucines (Ile52 and Ile263) that shifted upon hIRE1α cLD oligomerization (*Figure 6D,E*), suggesting that they report on the formation of hIRE1α cLD's putative oligomerization interface IF2[L]. Lys347, Lys349 and Lys351 are located in a region that was not resolved in hIRE1α cLD crystal structure, suggesting that these regions are contributing to the formation of the oligomerization interface in hIRE1α cLD.

We next threaded the sequence of hIRE1α cLD into the yeast crystal structure of the oligomeric state, which fulfilled the distance restraints imposed by the cross-links (*Figure 7B,C*). This structural model predicted an extensive interface formed by hIRE1α cLD oligomers that involves residues from parts of hIRE1 cLD that are not resolved in the crystal structure, as well as the incomplete β-propeller involved in the formation of the oligomerization interface in yeast Ire1 cLD (*Figure 7C*).

We used the predictive power of the structural model (hIRE1 cLD threaded into the yeast structure) to identify a patch of four hydrophobic residues WLLI (aa 359–362) suggested to contribute to the hIRE1α cLD oligomerization interface IF2[L] (*Figure 7C*, *Figure 7—figure supplement 2*). Assuming that these residues would be critical for oligomerization, we mutated them (WLLI[359-362] to GSGS[359-362]; 'IF2[L] mutant') and assessed whether the hIRE1α cLD IF2[L] mutant formed oligomers by AUC sedimentation velocity analysis. The experiments revealed that, at a concentration (50 µM) where wild type hIRE1α cLD readily forms oligomers, the hIRE1α cLD IF2[L] mutant sediment as a single dimeric peak, showing that the mutation prevents hIRE1α cLD oligomerization (*Figure 7D*).

## The peptide-induced allosteric switch remains intact in hIRE1α cLD IF2[L] mutant

To assess whether hIRE1α cLD IF2[L] mutant is functional, we tested peptide binding by fluorescent anisotropy experiments. The IF2[L] mutant bound MPZ1-N and MPZ1-N-2X peptide at similar affinities to the wild type protein (with $K_{1/2} = 5.4 \pm 1.4$ µM and $K_{1/2} = 0.95 \pm 0.4$ µM, respectively) (*Figure 7E*, *Figure 7—figure supplement 3A*). These results indicated that hIRE1α cLD dimer is the functional unit for peptide binding and that hIRE1α cLD oligomers do not display a higher affinity conformation. Moreover, they also showed that the avidity effect that resulted in higher affinity binding of MPZ1-N-2X peptide to hIRE1α cLD does not require formation of higher hIRE1α cLD oligomers. AUC data confirmed these analyses and showed that binding of MPZ1-N-2X to hIRE1α cLD IF2[L]mutant stabilized dimer formation but did not lead to formation of oligomers bridged by MPZ1-N-2X peptide (*Figure 7E*, *Figure 7—figure supplement 3B and C*).

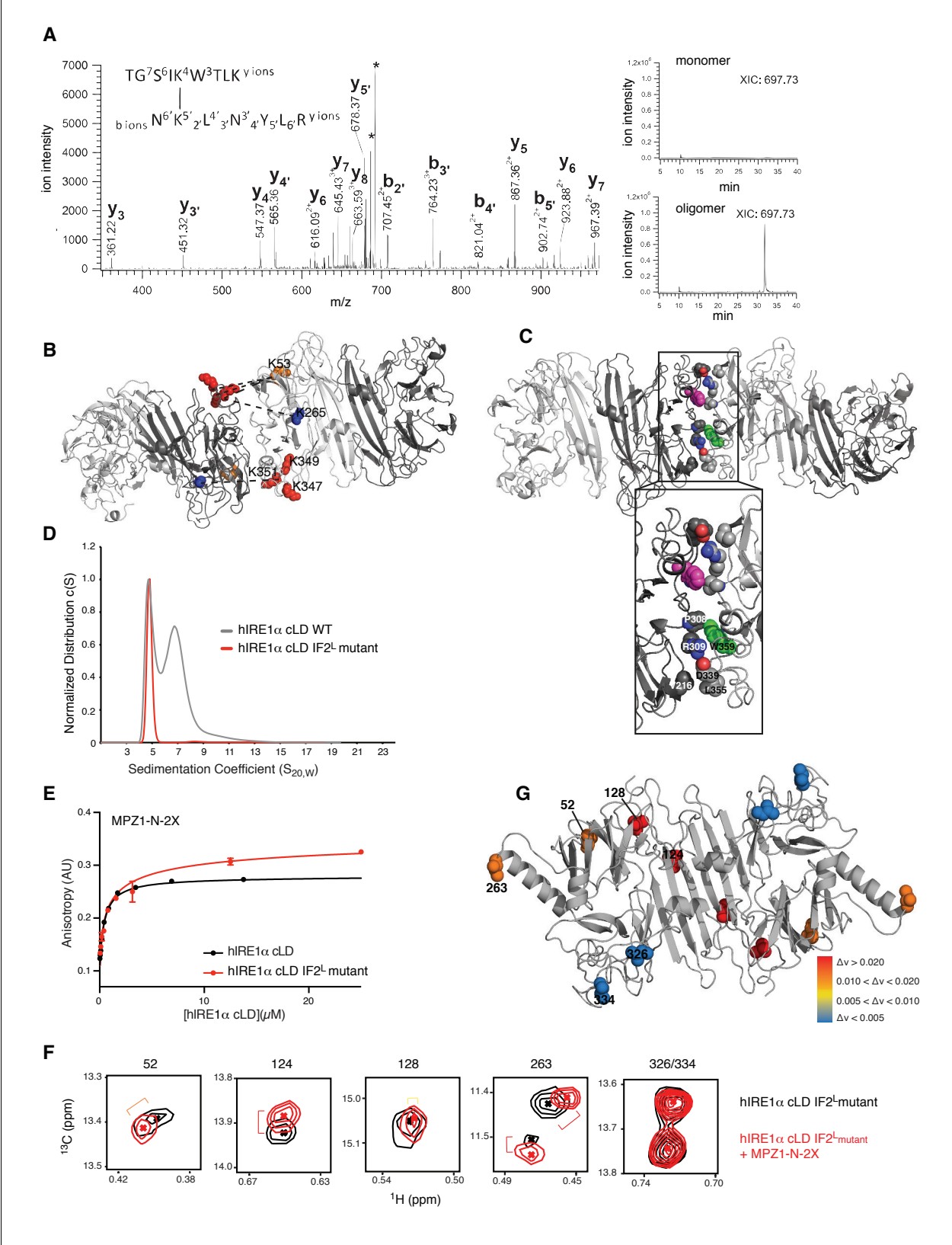

**Figure 7.** Cross-linking coupled to mass spectrometry identified the oligomerization interface of hIRE1α cLD  (**A**) Tandem mass spectrometry (MS) profile of the peptide crosslinked at Lys53 and Lys351. Extracted Ion chromatography (XIC) of the peptide peak in monomeric versus oligomeric hIRE1α

*Figure 7 continued on next page*

*Figure 7 continued*

cLD shows its absence in cross-linker treated monomer proteins. (**B**) Mapping cross-link sites on the structural model of hIRE1α cLD by threading on the oligomeric yeast crystal structure. Each monomer is colored as shades of gray. The Lys53, Lys265 are shown as orange and blue spheres, respectively, and Lys347,Lys349, Lys351 are shown as red spheres. The dashed lines indicate the cross-links between the lysines. (**C**) The amino acids forming the oligomerization interface are shown as spheres and colored by red (indicating oxygens), blue (indicating nitrogens) and white (indicating carbons). The Trp359 that is mutated in the hIRE1α cLD IF2$^L$ mutant is colored as green and pink in different protomers. (**D**) $^{359}$WLLI$^{362}$-GSGS mutation (hIRE1α cLD IF2$^L$ mutant) impairs hIRE1α cLD oligomerization determined by AUC sedimentation velocity experiments. Gray line depicts wild type hIRE1 α cLD (25 μM) and red shows the hIRE1α cLD IF2$^L$ mutant (50 μM). (**E**) hIRE1α cLD IF2$^L$ mutant binds MPZ1-N-2X peptide (red curve) at similar affinity K $_{1/2}$=0.95 ± 0.4 as wild type hIRE1α cLD (black curve)(K$_{1/2}$ = 0.456 ± 0.7 μM) determined by fluorescence anisotropy measurements. (**F**) Close-up of isoleucine peaks in the methyl-TROSY spectrum of hIRE1α cLD IF2$^L$ mutant alone (black, 50 μM) overlaid with the spectrum of hIRE1α cLD bound to MPZ1-N-2X (upper panel, red, 1:1 molar ratio). (**G**) The isoleucine peaks shifting upon peptide binding to hIRE1α cLD IF2$^L$ mutant are mapped on hIRE1α cLD structural model based on the chemical shift perturbations calculated in *Figure 7—figure supplement 4*) consistent with the color code in *Figure 4E*.

DOI: https://doi.org/10.7554/eLife.30700.021

The following figure supplements are available for figure 7:

**Figure supplement 1.** SDS-PAGE analysis of BS3 (1 mM) cross-linked hIRE1α cLD (20 μM).

DOI: https://doi.org/10.7554/eLife.30700.022

**Figure supplement 2.** The hydrophobic stretch $^{359}$WLLI$^{362}$ that is mutated in hIRE1α cLD IF2$^L$ mutant is shown on the structural model of hIRE1α cLD based on yeast crystal structure.

DOI: https://doi.org/10.7554/eLife.30700.023

**Figure supplement 3.** hIRE1α cLD IF2$^L$ mutant binds peptides.

DOI: https://doi.org/10.7554/eLife.30700.024

**Figure supplement 4.** Peptide induced allosteric coupling is intact in hIRE1α cLD IF2$^L$ mutant.

DOI: https://doi.org/10.7554/eLife.30700.025

The hIRE1α cLD IF2$^L$ mutant therefore enabled us to decouple peptide induced allosteric communication from the formation of oligomers, both of which could have contributed to the shift of the isoleucine peaks in the NMR experiments. To address this notion, we repeated the NMR experiments with the IF2$^L$ mutant (*Figure 7F*, *Figure 7—figure supplement 4A–C*). Similar to WT hIRE1α cLD, MPZ1-N peptide binding to hIRE1α cLD IF2$^L$ mutant shifted isoleucines in the β-sheet floor (marked by Ile124 and Ile128) (*Figure 7F and G*, *Figure 7—figure supplement 4B and C*). Importantly, isoleucine peaks (Ile52 and Ile263) close to the oligomerization interfaces also shifted upon peptide binding to the hIRE1α cLD IF2$^L$ mutant. Thus peptide binding-induced conformational changes in isoleucines distant to the peptide binding site persisted in the hIRE1α cLD IF2$^L$ mutant. Interestingly, MPZ1-N-2X binding shifted isoleucine peaks in the same direction and to a similar extent as binding of MPZ1-N, indicating that hIRE1α cLD IF2$^L$ binds to the same site in these

**Table 1.** List of cross-linked peptides in hIRE1α cLD detected by mass spectrometry and their relative abundance.

| Xlink 1 | Xlink 2 | | Band number on the SDS-PAGE (with peptides) | | | | | Xlink 1 | Xlink 2 | | Band number on the SDS-PAGE (without peptides) | | | | |
|---|---|---|---|---|---|---|---|---|---|---|---|---|---|---|---|
| | | | 5 | 4 | 3 | 2 | 1 (monomer) | | | | 5 | 4 | 3 | 2 | 1 (monomer) |
| K53 | K347 | AVE (%) | 4.1% | 3.2% | 3.7% | 4.9% | 0.0% | K53 | K347 | AVE (%) | 3.3% | 4.3% | 4.4% | 3.4% | 0.0% |
| | | STDEV(%) | 0.4% | 1.9% | 1.0% | 1.0% | 0.0% | | | STDEV(%) | 0.3% | 1.9% | 2.1% | 2.5% | 0.0% |
| K53 | K349 | AVE (%) | 1.4% | 1.0% | 1.4% | 1.5% | 0.0% | K53 | K349 | AVE (%) | 1.6% | 1.4% | 1.6% | 1.6% | 0.0% |
| | | STDEV(%) | 0.2% | 0.4% | 0.3% | 0.8% | 0.0% | | | STDEV(%) | 0.5% | 0.5% | 0.9% | 0.5% | 0.0% |
| K53 | K351 | AVE (%) | 4.0% | 2.4% | 3.2% | 3.4% | 0.0% | K53 | K351 | AVE (%) | 2.3% | 3.2% | 2.6% | 2.1% | 0.0% |
| | | STDEV(%) | 1.1% | 2.4% | 0.8% | 0.3% | 0.0% | | | STDEV(%) | 0.4% | 1.0% | 1.7% | 2.3% | 0.0% |
| K121 | K121 | AVE (%) | 3.7% | 2.9% | 1.5% | 0.8% | 0.0% | K121 | K121 | AVE (%) | 2.6% | 2.1% | 2.1% | 0.3% | 0.0% |
| | | STDEV(%) | 0.9% | 1.4% | 0.0% | 0.3% | 0.0% | | | STDEV(%) | 0.2% | 0.8% | 1.5% | 0.1% | 0.0% |
| K351 | K265 | AVE (%) | 1.2% | 0.8% | 1.0% | 0.4% | 0.0% | K351 | K265 | AVE (%) | 0.6% | 0.9% | 0.9% | 0.4% | 0.0% |
| | | STDEV(%) | 0.1% | 0.6% | 0.5% | 0.3% | 0.0% | | | STDEV(%) | 0.3% | 0.2% | 0.4% | 0.3% | 0.0% |
| K53 | K265 | AVE (%) | 0.5% | 0.4% | 0.2% | 0.2% | 0.0% | K53 | K265 | AVE (%) | 0.3% | 0.4% | 0.3% | 0.1% | 0.0% |
| | | STDEV(%) | 0.1% | 0.1% | 0.0% | 0.0% | 0.0% | | | STDEV(%) | 0.1% | 0.3% | 0.1% | 0.1% | 0.0% |

DOI: https://doi.org/10.7554/eLife.30700.026

peptides (*Figure 7G*, *Figure 7—figure supplement 4B and C*). These data suggest that the increased affinity of MPZ1-N-2X is due to a decreased rate of dissociation of the peptide.

## IRE1 lumenal domain-driven oligomerization is crucial for IRE1 function in mammalian cells

To test the importance of lumenal domain driven oligomerization for hIRE1α function in vivo, we generated cell lines that stably express hIRE1α IF2$^L$ mutant as the only form of hIRE1α. To this end, we introduced the hIRE1α IF2$^L$ mutant into mouse embryonic fibroblasts (MEFs) deficient for both isoforms of IRE1 (IRE1α$^{-/-}$/IRE1β$^{-/-}$). In addition, we attached a GFP tag to IRE1's cytoplasmic flexible linker retaining its function as published previously for HEK293 cells (*Li et al., 2010*). In parallel, we introduced hIRE1α-GFP to IRE1α$^{-/-}$/IRE1β$^{-/-}$ MEFs to compare hIRE1α activity at similar conditions. In these cell lines, we controlled hIRE1α expression via a doxycycline-inducible promoter. In the absence of doxycycline, cells expressed low levels of hIRE1α due to the leakiness of the promoter. In those conditions, the expression level of the hIRE1α-GFP-IF2$^L$ mutant was similar to hIRE1α-GFP and to the level of endogenous IRE1α from wild-type MEFs, as assessed by Western blot analysis (*Figure 8A,B*).

We next monitored the *XBP1* mRNA splicing activity of IRE1 in IRE1α$^{-/-}$/IRE1β$^{-/-}$ MEFs harboring hIRE1α-GFP or hIRE1α-GFP-IF2$^L$ mutant (*Figure 8C*). We found that unlike hIRE1α-GFP, hIRE1α-GFP-IF2$^L$ mutant did not splice *XBP1* mRNA after induction of ER stress by tunicamycin, a chemical stressor that impairs ER-folding homeostasis by inhibiting N-linked glycosylation (*Figure 8C*, *Figure 8—figure supplement 1*) (*Heifetz et al., 1979*). IRE1's RNase activity is preceded by the autophosphorylation of its kinase domain, which can be monitored by a phospho-specific antibody. Western blot analysis showed no signal corresponding to phospho-IRE1 in the IRE1α$^{-/-}$/IRE1β$^{-/-}$ cells expressing hIRE1α–GFP-IF2$^L$, by contrast to the same cells reconstituted with wild type hIRE1α-GFP, or in contrast to wild type MEFs, in which we detected phosphorylation of the endogenous protein (*Figure 8B*, *Figure 8—figure supplement 2*). Lastly, confocal microscopy revealed that under ER stress conditions where hIRE1α-GFP readily formed foci (>70%, n = 88, *Figure 8D*, *Figure 8—figure supplement 3A*), reflecting its assembly into active oligomers, the hIRE1α-GFP-IF2$^L$ mutant failed to do so (*Figure 8D*, *Figure 8—figure supplement 3B and C*). These data confirmed that cLD-mediated oligomerization is crucial for IRE1 function in cells.

## Discussion

To date, the mechanism by which mammalian IRE1 senses ER stress has remained controversial. Here, we provide evidence that activation of human IRE1α occurs via direct recognition of unfolded proteins and that the mechanism of ER stress sensing is conserved from yeast to mammals. This conclusion is based on six independent lines of evidence. First, we found that hIRE1α cLD binds peptides with a characteristic amino acid bias. Second, NMR spectroscopy suggested that peptides bind to hIRE1α cLD's MHC-like groove and induce a conformational change including the distant αB helix. In this way, occupation of the peptide-binding groove is allosterically communicated, which, we propose, culminates in the formation of a functional oligomerization interface corresponding to IF2$^L$ in yIRE1 cLD. Third, binding of minimal-length peptides induces formation of hIRE1α cLD oligomers as assessed by AUC analyses, further supporting this notion. Fourth, cross-linking experiments captured the oligomerization interfaces, which allowed identification of a functionally crucial hydrophobic patch at IF2$^L$. Fifth, mutation of this patch uncoupled peptide binding from oligomerization but retained the allosteric coupling within the domain. Sixth, impairing the oligomerization of hIRE1α cLD abolished IRE1's activity in living cells, attesting to the physiological relevance of the activation mechanism proposed here.

Taken together, our data converge on a model (*Figure 9*) in which unfolded protein-binding activates a switch in hIRE1α's cLD, leading to rearrangements that render it compatible with the formation of IF2$^L$ and therefore stabilizing an active oligomeric conformation (*Video 1*). cLD-mediated oligomerization on the lumenal side of the ER, in turn, would juxtapose hIRE1α's cytosolic kinase domains in the face-to-face confirmation allowing its *trans*-autophosphorylation, followed by stacking of its RNase domains in back-to-back orientation. These conformational rearrangements then lead to RNase activation, and thus allowing information flow across the ER membrane. Interestingly,

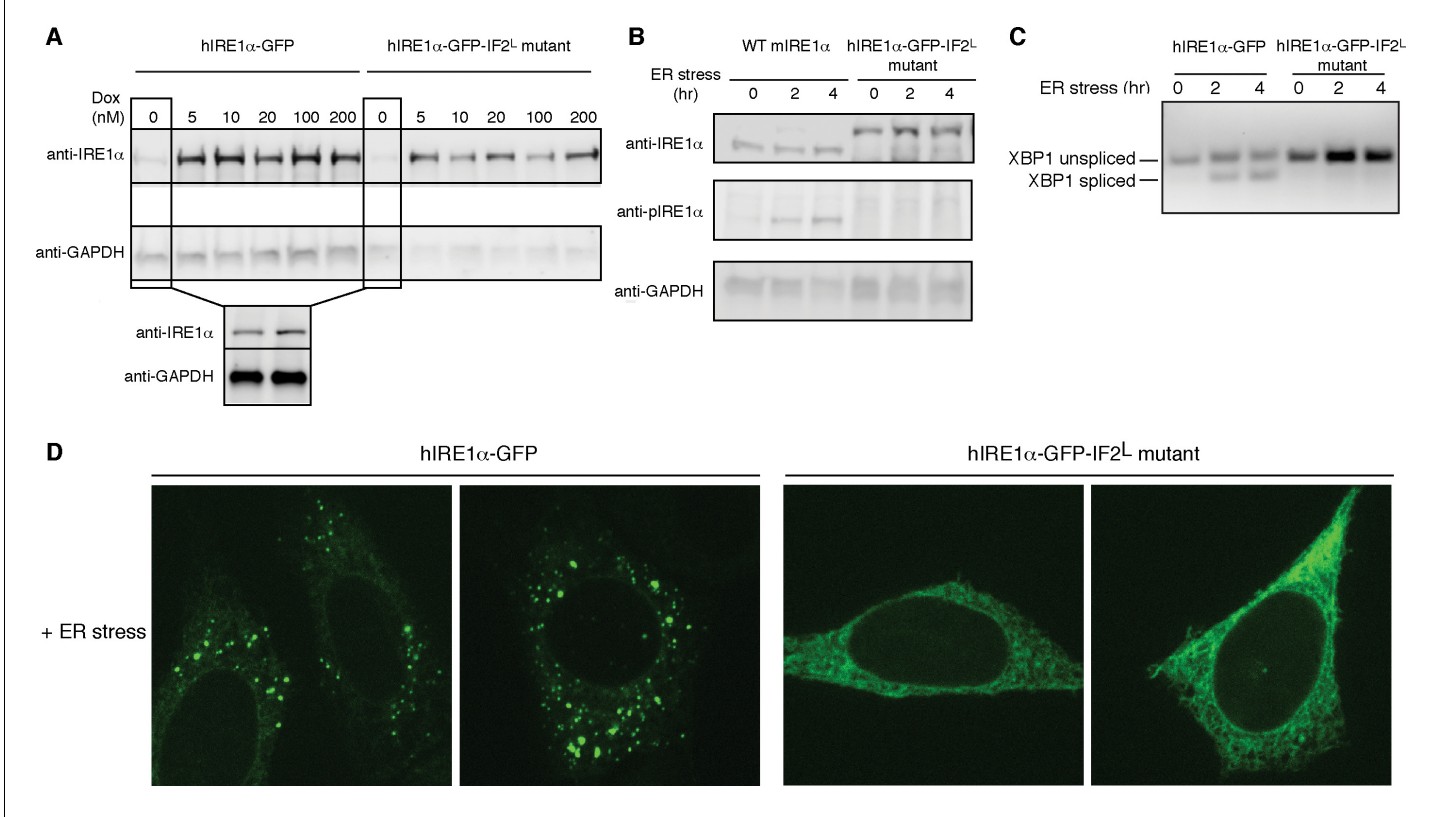

**Figure 8.** Lumenal domain driven oligomerization is crucial for IRE1 function (A) Western blot analyses show the levels of hIRE1α-GFP and hIRE1α-GFP-IF2$^L$ mutant proteins stably expressed in IRE1α$^{-/-}$/IRE1β$^{-/-}$ MEFs in response to various doxycycline concentrations. hIRE1α is detected by anti-IRE1 antibody and GAPDH is probed as the loading control. The lower panel shows Western blot analysis comparing IRE1 levels in hIRE1α-GFP and hIRE1α-GFP-IF2$^L$ mutant cell lines in the absence of doxycycline side by side. (B) Western blot analyses of hIRE1α-GFP and hIRE1α-GFP-IF2$^L$ mutant reconstituted in IRE1α$^{-/-}$/IRE1β$^{-/-}$ MEFs and MEFs isolated from wild type mice are probed with anti-IRE1 and anti-phospho-IRE1 antibody. The cells are treated with 5 µg/ml tunicamycin for inducing ER stress. (C) Unlike hIRE1α-GFP, the hIRE1α-GFP-IF2$^L$ mutant does not splice *XBP1* mRNA after induction of ER stress by the chemical ER stressor tunicamycin (5 µg/ml). *XBP1* mRNA splicing is determined by semi quantitative PCR. The spliced and unspliced forms of *XBP1* mRNA are indicated. Splicing assays in are conducted in IRE1α$^{-/-}$/IRE1β$^{-/-}$ MEFs reconstituted with hIRE1α-GFP or the hIRE1α-GFP-IF2$^L$ mutant in the absence of doxycycline. (D) Confocal microscopy images of IRE1α$^{-/-}$/IRE1β$^{-/-}$ MEFs reconstituted with hIRE1α-GFP-IF2$^L$ mutant and hIRE1α-GFP after 4 hr of chemically induced ER stress by tunicamycin (5 µg/ml).

DOI: https://doi.org/10.7554/eLife.30700.027

The following figure supplements are available for figure 8:

**Figure supplement 1.** Unlike hIRE1α-GFP, the hIRE1α-GFP-IF2$^L$ mutant does not splice *XBP1* mRNA after induction of ER stress by tunicamycin (Tm, 5 µg/ml) at various time points and 2 hr after thapsigargin (Tg, 100 nM) treatment.

DOI: https://doi.org/10.7554/eLife.30700.028

**Figure supplement 2.** Western blot analyses of cell lysates collected at various times after ER stress induction from IRE1α−/−/IRE1β−/− MEFs reconstituted with hIRE1α-GFP and hIRE1α-GFP-IF2$^L$ ('L' for lumenal) mutant are probed with anti-phospho IRE1 antibody (upper panel), whereas GAPDH is the loading control (lower panel).

DOI: https://doi.org/10.7554/eLife.30700.029

**Figure supplement 3.** hIRE1α-GFP-IF2$^L$ mutant does not form foci.

DOI: https://doi.org/10.7554/eLife.30700.030

our data show that impairment of lumenal domain oligomerization diminished IRE1's both RNase and kinase activities in cells.

Currently due to lack of biochemical and structural understanding of IRE1's interaction with the ER-resident chaperone BiP, its role in regulating IRE1 activity remains unknown. Although it is clear that BiP is released from IRE1 upon ER stress (*Bertolotti et al., 2000*), current models proposing BiP as the primary regulator of IRE1 activity do not address how active IRE1 oligomers would form (*Carrara et al., 2015*; *Oikawa et al., 2009*; *Zhou et al., 2006*). By contrast, our data indicate that

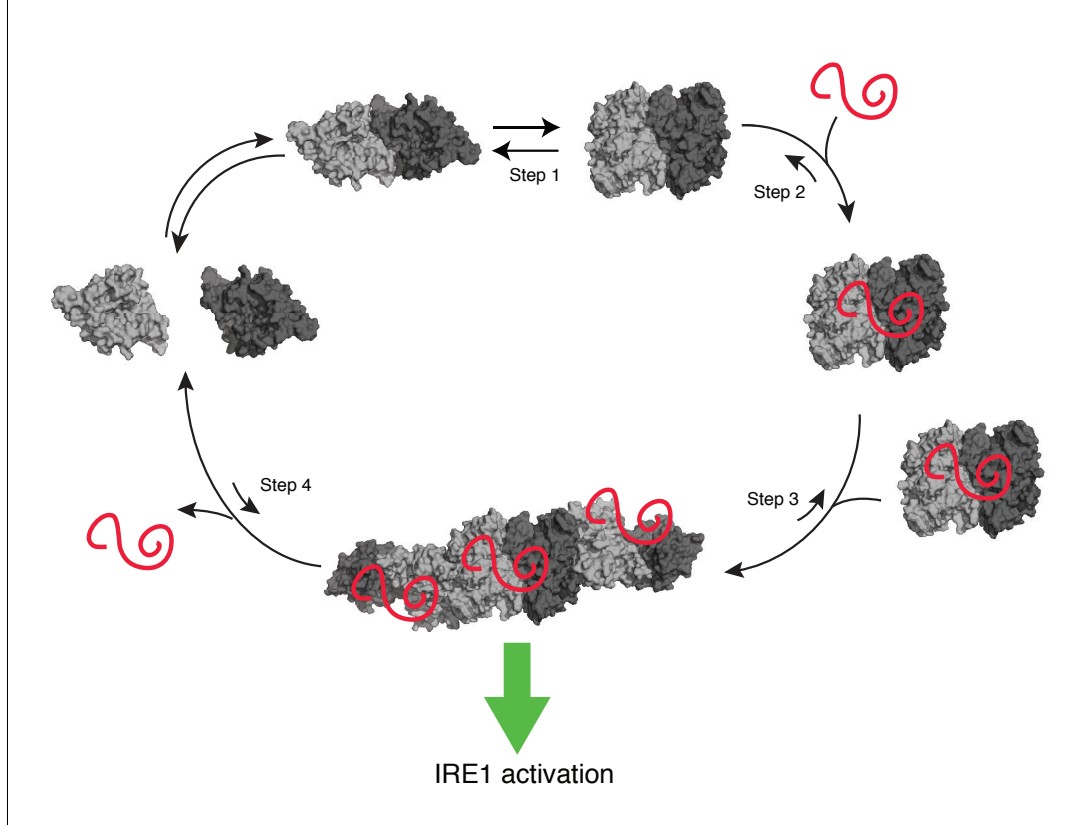

**Figure 9.** Model of human IRE1 activation  Apo-hIRE1α LD dimers are found in equilibrium between closed and open conformations (Step 1) (note that for simplicity IRE1's cytoplasmic kinase/RNase domains are not displayed in the model). Upon ER stress, unfolded proteins accumulating in the ER lumen bind hIRE1α LD. Unfolded protein binding stabilizes hIRE1α LD in the open conformation and induces a conformational change in the αB helix and the neighboring structural elements (Steps 2). This conformational change releases the block on oligomerization, thus leading to active hIRE1α oligomers by allowing the formation of an IF2$^L$-like interface in hIRE1α LD (Step 3). Oligomerization driven by hIRE1α LD subsequently activates its kinase and RNase domains. When protein-folding homeostasis is achieved, the dynamic hIRE1α LD oligomers re-adopt the inactive hIRE1α LD conformation (Step 4).

DOI: https://doi.org/10.7554/eLife.30700.031

peptide-binding is important for lumenal domain-driven IRE1 oligomerization, leading to its activation. We therefore consider it most plausible that BiP binding modulates the response via tuning IRE1's oligomerization equilibrium, similar to what was shown for the yeast counterpart (*Pincus et al., 2010*). In this way, BiP binding would buffer IRE1 activity at the early stages of the ER stress when the chaperones are not overwhelmed by the unfolded protein load, and during the deactivation phase, when the protein folding homeostasis is achieved. In this scenario, unfolded protein accumulation exerts synergistic effects on IRE1 activation, simultaneously freeing more IRE1 from BiP upon ER stress and inducing IRE1's oligomerization/activation through their direct binding to the sensor (*Pincus et al., 2010*).

Despite these profound similarities in the salient features of ER stress sensing and processing, yeast and human IRE1α cLD display some distinct oligomerization properties. Whereas yIRE1 cLD precipitously assembles into larger oligomers at concentrations that exceed its dimerization constant (*Gardner and Walter, 2011*), hIRE1α cLD forms discrete dimers, which in a concentration-dependent manner gradually assemble into tetramers. hIRE1α cLD oligomers are in a dynamic equilibrium of different states, apparent from our size exclusion chromatography and AUC analyses and hIRE1α cLD forms even larger oligomers when bound to peptides. These observations are consistent with the model that the αB helix, which may hinder formation of hIRE1α oligomers as previously suggested (*Zhou et al., 2006*) participates in conformational changes that release its block on oligomerization. At higher hIRE1α cLD concentrations, the conformational equilibrium of the αB helix is

shifted towards the active state. Peptide binding allosterically releases this inhibition and stabilizes the active hIRE1$\alpha$ oligomers. We anticipate that the effect of peptide binding-induced oligomerization would be more pronounced under physiological conditions, where hIRE1$\alpha$ is tethered to ER-membrane with diffusion limited to two dimensions.

We speculate that the conformational change in the $\alpha$B helix allows the incomplete $\beta$-propeller to form contacts with the residues from the flexible region, which is not resolved in the crystal structure (V307-Y358) forming the oligomerization interface in hIRE1$\alpha$ cLD. In this conformation, $\alpha$B helix may provide additional contact sites contributing to the oligomerization interface. Interestingly, one of the symmetry mates captured by hIRE1$\alpha$ cLD crystal structure shows contacts of the $\alpha$B helix with the hydrophobic stretch ($^{359}$WLLI$^{362}$), which we show to be important for oligomerization. We anticipate that in addition to this hydrophobic stretch, additional contacts contributed by these flexible parts may further facilitate oligomer formation.

hIRE1$\alpha$ cLD's groove is enriched in aromatic residues and displays a negatively charged surface. In this way, the amino acids lining the groove chemically complement hIRE1$\alpha$ cLD binding peptides identified in our study, which are enriched in aromatics and arginines. In the crystal structure of hIRE1$\alpha$ cLD in the 'closed' conformation, the $\alpha$-helices forming the MHC-like groove are close together and mask the residues on the $\beta$-sheet floor. When these helices are moved approximately 6 Å apart from one another, the groove deepens and exposes more hydrophobicity mostly contributed by newly exposed aromatic residues. Thus, opening the groove exposes surface chemistry that is conducive to IRE1 binding peptides. Our data support a model in which widening of the groove is allosterically coupled to the formation of the IF2$^L$-like oligomerization interfaces.

We showed that a 12-mer peptide is the shortest derivative of MPZ1 peptide that binds hIRE1$\alpha$ cLD with undiminished affinity when compared to the original 21-mer peptide, indicating that a 12-mer provides maximal contact with cognate interfaces in hIRE1$\alpha$ cLD groove. It is plausible that similar to MHC molecules, select amino acids in unfolded polypeptides act as 'anchor residues' providing contact sites for hIRE1 $\alpha$ cLD binding (*Fremont et al., 1992*; *Matsumura et al., 1992*; *Wilson and Fremont, 1993*). Notably, assuming an extended peptide backbone with an average length of 3.4 Å per peptide bond, a 12-mer peptide can fit without constraints into the 39 Å-long groove in the structural model presented here. This notion suggests that the groove ensures preferential binding of fully exposed, unfolded 39Å-stretch of a polypeptide chain. This recognition principle is therefore similar to that of Hsp70-type chaperones, where the structural constraints in the cavity of the substrate-binding domain allow interaction with the substrates only in their extended, unfolded conformation, although Hsp70 only binds a seven amino acids stretch (*Rüdiger et al., 1997a*; *Rüdiger et al., 1997b*).

Supporting the notion of mechanistic similarities in unfolded protein recognition between chaperone proteins and IRE1, hIRE1$\alpha$ cLD and the ER-resident chaperone BiP bind partially overlapping as well as distinct sets of peptides tested in our peptide arrays, as previously shown for the orthologous yeast proteins (*Gardner and Walter, 2011*). Importantly, the presence of distinct hIRE1$\alpha$ cLD binding peptides liberates IRE1 from an otherwise inevitable failure to compete with highly abundant BiP for binding sites in unfolded proteins. hIRE1$\alpha$ cLD's affinity for peptides measured here varied between 5 and 30 µM, which is within the same order of magnitude but at the lower range of those reported for most chaperones (*Karagöz et al., 2014*; *Marcinowski et al., 2011*; *Street et al., 2011*). For example, hIRE1$\alpha$ cLD binds the IgG's C$_H$1 unfolded domain with ~30 µM affinity whereas BiP was shown to bind the same protein with ~7 µM (*Marcinowski et al., 2011*). We surmise that this difference has been selected in evolution to set the threshold for unfolded protein recognition slightly higher for the UPR sensors when compared to that of chaperones so that the UPR is not triggered until a critical concentration of unfolded proteins accumulates. Moreover, our data with the MPZ1-N-2X peptide suggested that IRE1 could display higher affinity for select polypeptides that present more than a single IRE1 binding site.

IRE1 dysfunction contributes to the development of numerous diseases, including cancer (such as multiple myeloma [*Mimura et al., 2012*]), metabolic disorders (such as obesity and diabetes [*Fonseca et al., 2009*; *Hotamisligil, 2010*]) and neurodegenerative diseases (such as amyotrophic lateral sclerosis and Hungtinton's disease [*Hetz et al., 2009*; *Matus et al., 2009*; *Vidal et al., 2012*]). Depending on the disease context, IRE1 makes life or death decisions in response to altered ER function manifested in these pathological conditions (Walter P. and D., 2011). Our data showing that unfolded proteins stabilize a distinct IRE1 conformation suggest novel approaches to

manipulate IRE1 pharmacologically. For example, it will be promising to design or screen for small molecule modulators that lock IRE1's groove in the open or closed conformation based on the chemical signature of the IRE1 binding peptides identified here. Such compounds could act as agonists or antagonists of IRE1 activity. As such, it should be possible to develop new classes of pharmaceuticals to induce or inhibit the IRE1 branch of the UPR, driving the desired IRE1 output depending on the disease context.

## Materials and methods

### Reagents

Synthetic peptides were ordered from Elim Biosciences and GenScript at >95% purity.

### Protein purification

To express MBP-hIRE1$\alpha$ cLD (aa 24–389), human IRE1$\alpha$ cDNA sequences were cloned into a pMalC2p vector to create a hIRE1$\alpha$ cLD fused on its N-terminus to MBP. To express His$_{10}$-hIRE1$\alpha$ cLD, hIRE1$\alpha$ cLD was cloned into pet16b(+) vector containing a FactorXa protease cleavage site. Additionally, His$_{10}$-hIRE1$\alpha$ cLD and IRE1 LD coding sequences were cloned into pet47b(+) vector with a preScission protease cleavage site. Hamster BiP with an N-terminal His$_{10}$-tag was cloned into pet16b(+) vector, which was modified to introduce a preScission protease site C-terminal to the His$_{10}$-tag. For expression of the proteins, the plasmid of interest was transformed into *Escherichia coli* strain BL21DE3* RIPL (Agilent Technologies) or Rosetta2 cells (Novagen). Cells were grown in Luria Broth at 37°C until $OD_{600}$ = 0.6. Protein expression was induced with 0.3 mM IPTG, and cells were grown at 21°C overnight. For selective labeling, cells were grown according to published protocols (*Tugarinov and Kay, 2004*). Briefly, cells were grown at minimal media in $D_2O$ supplemented with deuterated glucose as the primary carbon source. For purification, cells were resuspended in Lysis Buffer (50 mM HEPES pH 7.2, 400 mM NaCl, 4 mM dithiothreitol (DTT)(or 5 mM β-mercaptoethanol, if a nickel column was used)) and were lysed in an Avestin EmulsiFlex-C3 cell disruptor at 16,000 psi. The supernatant was collected after centrifugation for 40 min at 30,000xg. MBP-IRE1 cLD constructs were purified on an MBP-amylose resin (New England Biolabs) and eluted with 10 mM amylose in Elution Buffer (50 mM HEPES pH 7.2, 150 mM NaCl, 4 mM DTT) after washing the column with 20 column volumes of Lysis Buffer. The eluate was then diluted with 50 mM HEPES (pH 7.2) buffer to 50 mM NaCl and applied to a MonoQ ion exchange column and eluted with a linear gradient from 50 mM to 1 M NaCl. The protein was further purified on a Superdex 200 10/300 gel filtration column equilibrated with Buffer A (25 mM HEPES pH 7.2, 150 mM NaCl, 2 mM *tris*(2carboxyethyl)phosphine (TCEP). The initial purification of His$_6$- and His$_{10}$-hIRE1$\alpha$ cLD and His$_{10}$-BiP constructs were performed on a His-TRAP column (GE Healthcare), where the protein was eluted with gradient from 20 mM to 500 mM imidazole. The eluate was purified on a MonoQ column, before the His$_6$-tag (pet47b+) or His$_{10}$-tag (pet16b+) were removed by either PreScission protease (GE Healthcare, 1 unit of enzyme for 100 μg of protein) or FactorXa (NEB, 1 μg of FactorXa per 100 μg of protein), respectively. The tag removal was performed at 4° C overnight after the protein concentration was adjusted to 1 mg/mL. $C_{H1}$ domain of IgG was purified under reducing conditions as described (*Feige et al., 2009*). Protein concentrations were determined using extinction coefficient at 280 nm predicted by the Expasy ProtParam tool (http://web.expasy.org/protparam/).

### Peptide arrays

Peptide arrays were purchased from the MIT Biopolymers Laboratory. The tiling arrays were composed of 18-mer peptides that were tiled along the CPY*, MPZ, insulin, lysozyme and PTIP sequences with a three amino acid shift between adjacent spots. In the mutational arrays, peptides were synthesized to systematically mutate each amino acid in the core region of the CPY*-derived peptide. The arrays were incubated in 100% methanol for 10 min, then in Binding Buffer (50 mM HEPES pH 7.2, 150 mM NaCl, 0.02% Tween-20, 2 mM DTT) three times for 10 min each. For BiP experiments, ADP and $MgCl_2$ were added to the binding buffer to final concentrations of 1 mM and 5 mM, respectively. The arrays were then incubated for 1 hr at room temperature with 500 nM MBP-hIRE1$\alpha$ cLD or His$_{10}$-BiP and washed again three times with 10 min incubation in between the washes in Binding Buffer to remove any unbound protein. Using a semi-dry transfer apparatus, the

bound protein was transferred to a nitrocellulose membrane and detected with anti-MBP antiserum (NEB) or anti-His$_6$ antibody (Abcam). The contribution of each amino acid to hIRE1$\alpha$ cLD and BiP binding was calculated as described previously (*Gardner and Walter, 2011*). The peptide arrays were quantified using Max Quant. The binding intensity in each spot was normalized to max signal intensity in the peptide array. The peptides with the top 10% binding scores were selected and the occurrence of each amino acid in these top-binding peptides was calculated. This value is normalized to their abundance in the arrays (*Figure 2A*). To calculate experimental error, the amino acid occurrences of top binders were calculated for independent replicates. The statistical significance (p<0.05) is determined using non-paired t-test by the Prism software (*Figure 2—figure supplement 1A*).

## Fluorescence anisotropy

For fluorescence anisotropy measurements, MPZ1 peptide attached to 5-carboxyfluorescein (5-FAM) at its C-terminus was obtained at >95% purity from ELIM Biopharmaceuticals. For the remaining peptides (8ab1, MPZ1-N, MPZ1-M, MPZ1-C and MPZ1-N) derivatives were synthesized with 5-FAM attached to their N-terminus by GenScript at >95% purity. Binding affinities of hIRE1$\alpha$ cLD or IRE1 mutants to FAM-labeled peptides were measured by the change in fluorescence anisotropy on a Spectramax-M5 plate reader with excitation at 485 nm and emission at 525 nm with increasing concentrations of hIRE1$\alpha$ cLD. Fluorescently labeled peptides were used in a concentration range of 50–100 nM. The reaction volume of each data point was 20 µL and the measurements were performed in 384-well, black flat-bottomed plates after incubation of peptide with hIRE1$\alpha$ cLD or its mutants for 30 min at 25° C. Binding curves were fitted using Prism Software (GraphPad) using the following equation: $F_{bound} = r_{free} + (r_{max} - r_{free})/(1 + 10((LogK^{1/2}-X) \cdot n_H))$, where $F_{bound}$ is the fraction of peptide bound, $r_{max}$ and $r_{free}$ are the anisotropy values at maximum and minimum plateaus, respectively. $n_H$ is the Hill coefficient and x is the concentration of the protein in log scale. Curvefitting was performed with minimal constraints to obtain $K_{1/2}$ values with high $R^2$ values. However, as this equation does not take into account the equilibria between hIRE1$\alpha$ cLD dimers/oligomers, these apperant $K_{1/2}$ values do not reflect the dissociation constant.

## Microscale thermophoresis experiments (MST)

MST experiments were performed with a Monolith NT.115 instrument (NanoTemper Technologies, Germany). All experiments were done with the following buffer: 25 mM HEPES pH 7.2, 150 mM NaCl, 1 mM TCEP, 0.025% Tween-20. hIRE1$\alpha$ cLD was labeled using the Monolith NT Protein labeling Kit Red-Maleimide. Labeled protein was used in the measurements at a concentration of 50 nM. It was mixed with equal volumes unlabeled interaction partner in two-fold serial dilutions. Hydrophilic-treated capillaries (NanoTemper Technologies) were used for all the measurements. All experiments were performed at 50% LED power and 40-60–80% IR-laser at 25°C.

## AUC sedimentation velocity experiments

Sedimentation velocity experiments were carried out in a Beckman Optima XL-A analytical centrifuge at 40,000xg at 20°C with An-60 Ti rotor. All experiments were performed in buffer containing 25 mM HEPES pH 7.2, 150 mM NaCl, 2 mM DTT. Samples (400 µL) and reference buffer (410 µL) were loaded into AUC cells for each experiment. Samples of hIRE1$\alpha$ cLD at 5 µM were scanned at 280 nm, whereas hIRE1$\alpha$ cLD at concentrations higher than 25 µM were scanned at 290 nm to prevent detector saturation at high protein concentrations. Data analysis was performed using the SED-FIT software employing the c(s) method with time invariant and radial invariant noise fitting (*Schuck, 2000*). Buffer viscosity was calculated by Sednterp.

## NMR experiments

NMR experiments were performed on an 800 MHz Bruker AVANCE-I spectrometer with a TXI Cryo-probe equipped with an actively shielded Z-gradient at 298.0 K. Samples were buffer-exchanged into 25 mM phosphate buffer pH 7.2, 150 mM NaCl and 2 mM DTT in 100% D$_2$O on Vivaspin columns (Millipore). The concentration of WT hIRE1$\alpha$ cLD and hIRE1$\alpha$ cLD mutants varied from 25 to 400 µM in 250 µL volume. Samples were placed in a Shigemi advanced NMR microtube. For peptide and unfolded protein binding experiments, the peptides were dissolved in the same buffer at high

concentrations (1–2 mM) and titrated in 1:0.5, 1:1 and 1:2 molar ratios. Two-dimensional [$^{13}$C, $^{1}$H]-HMQC methyl correlation experiments on $^{13}$CH$_3$–Ile hIRE1α cLD were acquired with 86* and 768* complex points in the $^{13}$C and $^{1}$H dimensions, respectively. All spectra were processed with TOP-SPIN 3.2 and analyzed with Sparky.

## Attachment of the spin label to MPZ1 peptide and PRE experiments

MPZ1 peptide at 200 μM was labeled with 3-(2-iodoacetamido)-proxyl (Sigma) at the single cysteine, Cys5 in 25 mM phosphate buffer pH 7.2, 150 mM NaCl in the presence of 2 mM spin-label at 4°C for 8 hr. The labeled peptide was then dialyzed in a Slide-a-Lyzer dialysis cassette (Thermo Fisher Scientific) with 2 kDa cut-off to remove the excess spin-label and to exchange the buffer to deuterated buffer (25 mM phosphate buffer pH 7.2, 150 mM NaCl, 2 mM DTT) for NMR experiments. Control samples used in the reference experiments contained (1-oxyl-2,2,5,5-tetramethylpyrroline-3-methyl) methanethiosulfonate spin-label that was treated the same way as the proxyl-labeled peptide. Wild type hIRE1α cLD and quadruple mutant hIRE1α cLD (Leu186Ile, Ile326/334/362Val) and single mutant Leu186Ile were used in PRE experiments at 75 μM and 100 μM protein concentration respectively, in the presence and absence of equimolar concentrations of MPZ1-proxyl peptide. We normalized the PRE effect with the surface exposed area displayed by that isoleucine to exclude possible contributions from non-specific interactions with the spin label attached peptide (*Clore and Iwahara, 2009*). The normalized PRE values are calculated as follows, the solvent accessible surface area for isoleucines are calculated using the 'GETAREA' webserver (http://curie.utmb.edu/getarea.html, [*Fraczkiewicz and Braun, 1998*]) based on hIRE1α cLD crystal structure. The maximum solvent accessible surface by these isoleucines is normalized to one and the normalized values are multiplied with the PRE effect. The PRE effect is calculated by dividing the intensity of isoleucine signals in the control experiments with the isoleucine signals in the presence of MPZ1-proxyl peptide.

## Cross-linking experiments

10 μM, 20 μM and 50 μM hIRE1α cLD was incubated with 500 μM and 1 mM BS3 cross-linker for 15 and 30 min at room temperature. Same reaction was performed for hIRE1 cLD pre-bound to 50 μM MPZ1-N for 30 min on ice. The reaction was stopped with the addition of 1M Tris-HCl at pH 8.0 at end concentration of 50 mM Tris-HCl, and incubated for 10 min at room temperature before running the SDS-PAGE gel.

## LC-MS/MS analysis and cross-linked peptide identification

Cross-linked products were in-gel digested and analyzed by LC-MS and LC-MS-MS as described previously (*Wu et al., 2013*; *Zeng-Elmore et al., 2014*). Briefly, 1 μl aliquot of the digestion mixture was injected into an Dionex Ultimate 3000 RSLCnano UHPLC system (Dionex Corporation, Sunnyvale, CA), and separated by a 75 μm × 25 cm PepMap RSLC column (100 Å, 2 μm) at a flow rate of ~450 nl/min. The eluant was connected directly to a nanoelectrospray ionization source of an LTQ Orbitrap XL mass spectrometer (Thermo Scientific, Waltham, MA). LC-MS data were acquired in an information-dependent acquisition mode, cycling between a MS scan (m/z 315–2,000) acquired in the Orbitrap, followed by low-energy CID analysis on three most intense multiply charged precursors acquired in the linear ion trap.

Cross-linked peptides were identified using an integrated module in Protein Prospector, based on a bioinformatic strategy described previously (*Chu et al., 2010*; *Trnka et al., 2014*). The score of a cross-linked peptide was based on number and types of fragment ions identified, as well as the sequence and charge state of the cross-linked peptide. Only results where the score difference is greater than 0 (i.e. the cross-linked peptide match was better than a single peptide match alone) are considered. Tandem MS spectra of cross-linked peptides were manually inspected to ensure data quality. With the threshold of peptide score and expectation value for oligomer-only cross-linked peptides, no decoy match was returned.

## Lentiviral constructs and transduction

The coding sequence of wild type GFP-tagged IRE1 (*Li et al., 2010*) was amplified by PCR with Phsuion polymerase (NEB) and oligonucleotides with engineered restriction sites. The PCR product was introduced into the Gateway entry vector pSHUTTLE-CMV-TO (kind gift of A. Ashkenazi,

Genentech and (*Gray et al., 2007*) atcognate KpnI and EcoRI sites. The hIRE1α-GFP- IF2$^L$ mutant was generated in pSHUTTLE-CMV-TO by site directed mutagenesis of the wild-type sequence. The resulting clones were recombined into pGpHUSH.puro (kind gift of A. Ashkenazi, Genentech and [*Gray et al., 2007*]), a single lentivirus expression vector that allows the doxycyline-regulatable (Tet$^{ON}$) expression of a gene-of-interest. VSV-G pseudotyped lentiviral particles were prepared using standard protocols using 293METR packaging cells (kind gift of Brian Ravinovich, formerly at MD Anderson Cancer Center, [*Rabinovich et al., 2006*]). Viral supernatants were concentrated by filtration (Amicon Ultra centrifugal filter device, 100 kDa MWCO) and used to infect target cells by centrifugal inoculation (spinoculation) at 2000 rpm inn a Beckman GH3.8 rotor outfitted with plate carriers for 90 min in presence of 8 ug/mL polybrene. The cells were left to recover overnight following infection and were then subjected to puromycin selection as described below.

## Cell culture and generation of stable cell lines

IRE1 double-knockout Mouse Embryonic Fibroblasts (MEF) (IRE1α$^{-/-}$/IRE1β$^{-/-}$) and wild-type MEFs (kind gift of D. Ron, University of Cambridge). were grown in DMEM supplemented with 10% fetal bovine serum, 2 mM L-glutamine, and penicillin/streptomycin. Cells were not tested for the mycoplasma contamination. Lentiviral-transduced cells were selected with 6 µg/mL puromycin for 72 hr based on the puromycin concentration defined by the kill curve. Subsequently, a pulse of 25 nM doxycycline was given to induce expression of the GFP-tagged IRE1 transgenes for 10–12 hr. The following day, the doxycycline was washed out and pseudoclonal cell populations were selected by fluorescent activated cell sorting based on GFP expression for both wild-type (hIRE1α-GFP) and IF2$^L$ mutant (hIRE1α-GFP-IF2$^L$ mutant) forms of IRE1. The cells were selected in a FACS Aria instrument (BD FACSAria3), gating for a very narrow GFP expressing population. This procedure ensures selection of a pseudoclonal population where most cells have similar levels of expression of the transgene of interest while avoiding typical problems associated with monoclonal selection of IRE1-expressing cells; namely an aberrant UPR. The pseudoclonal populations were expanded and frozen as source stocks for experiments.

## Live cell imaging of hIRE1α -GFP and hIRE1α-GFP-IF2$^L$ mutant

IRE1 double-knockout MEFs (IRE1α$^{-/-}$/IRE1β$^{-/-}$) reconstituted with of hIRE1α -GFP or hIRE1α-GFP-IF2$^L$ mutant were split 2 days before imaging onto ibiTreat dishes (ibidi) at $5 \times 10^4$ cells/dish. 25 nM Doxyccline containing medium was added for 10–12 hr, withdrawn before imaging and replaced with imaging media consisting of Fluorobrite DMEM (Thermo Scientific), 2.5% FBS, and 5 mM Hepes at a pH of 7.0 . Cells were imaged at 37°C on a spinning disk confocal with Yokogawa CSUX A1 scan head, Andor iXon EMCCD camera and 40x Plan Apo air Objective NA 0.95 with a 1.5x tube lens for additional magnification giving 60x final or 100X objective. Images were acquired using 488 nm laser at a rate of one frame per 3 min with 300 ms exposure time for each time point for an hour. Images were collected after different time points following induction of ER stress by tunicamycin (5 µg/mL) or thapsigargin (100 nM).

## Immunofluorescence of hIRE1α -GFP and hIRE1α-GFP-IF2$^L$ mutant

IRE1 double-knockout MEFs (IRE1α$^{-/-}$/IRE1β$^{-/-}$) reconstituted with of hIRE1α -GFP and hIRE1α-GFP-IF2$^L$ mutant were grown similar to live cell imaging experiments. After stress induction at various time points, cells were washed three times with PBS followed by 3 min fixation with 100% methanol, and a three subsequent 5 min washes with PBS. As these fixation conditions kept GFP intact, immunostaining of hIRE1α for fluorescence imaging was not required. DAPI staining is performed according to manufacturer's instructions (Thermo Fisher).

## Generation of cDNA and semi-quantitative PCR

Cells exposed to DMSO or thapsigargin (100 nM) or tunicamycin (5 µg/ml) were collected in 0.5 ml of TRIzol reagent (Life Technologies) from a six well dish and total RNA was extracted following the manufacturer's recommendations. To generate cDNAs, 500 ng of total RNA were reverse transcribed using the SuperScript VILO system (Life Technologies) following the manufacturer's recommendations. The resulting 20 µl reverse transcription reactions were diluted to 10 times to 200 µl with 10 mM Tris– HCl pH 8.2, and 1% of this dilution was used for multiplex semiquantitative PCR.

The multiplex PCR was set up using 1 µM of the forward reverese primers, 0.4 units of Taq DNA polymerase (Thermo Scientific), 0.2 mM of each dNTP, and 1.5 mM MgCl2, in a 20 µl reaction using the following buffer system: 75 mM Tris–HCl pH 8.8, 20 mM (NH4)SO4, and 0.01% Tween-20. The oligonucleotide sequences are the following: Hs_XBP1_Fwd: 50 -GGAGTT AAGACAGCGCTTGG-30; Hs_XBP1_Rev: 50 -ACTGGGTCCAAGTTG TCCAG-30. The PCR products were amplified for 28 cycles and resolved on 3% agarose gels (1:1 mixture of regular and low-melting point agarose) stained with ethidium bromide.

### Protein analysis by Western-Blot

Cells were lysed in SDS-PAGE loading buffer (1% SDS, 62.5 mM Tris-HCl pH 6.8, 10% glycerol). Lysates were sonicated and equal amounts were loaded on SDS-PAGE gels (BioRad, Hercules, CA). Proteins were transferred onto nitrocellulose membranes and probed with primary antibodies diluted in Phosphate-buffered saline supplemented with 0.1% Tween 20% and 5% bovine serum albumin at 4°C, overnight. The following antibodies were used: anti-IRE1 (1:1000) (14C10, Cell Signaling Technology, Danvers, MA), anti-GAPDH (1:1000) (14C10, Cell Signaling Technology, Danvers, MA and anti-phosho IRE1 antibody(1:500). IRE1 anti-phospho antibody is a kind gift of Avi Ashkenazi's group at Genentech. An HRP-conjugated secondary antibody (Amersham, Piscataway, NJ) was employed to detect immunereactive bands using enhanced chemiluminescence (SuperSignal; Thermo Scientific, Waltham, MA) detected by Li-Cor instrument (Li-Core Biosciences).

## Acknowledgements

We thank the Walter lab members for insightful comments on the manuscript. Purified Δ131Δ protein was a generous gift from Daniel Elnatan and David Agard. We thank Johannes Buchner for $C_{H1}$ expression plasmid and Linda Hendershot for hamster BiP expression plasmid. We are thankful to Avi Ashkenazi and David Lawrence for the anti-phosho-IRE1 antibody and pGpHUSH.puro vector. We thank the MIT Biopolymers Institute for synthesizing the peptide arrays. This work was supported by NSF CLF #1307367 (to FC). PW is an Investigator of the Howard Hughes Medical Institute.

## Additional information

### Funding

| Funder | Grant reference number | Author |
| --- | --- | --- |
| National Science Foundation | CLF #1307367 | F Chu |
| Howard Hughes Medical Institute | | Peter Walter |

The funders had no role in study design, data collection and interpretation, or the decision to submit the work for publication.

### Author contributions

G Elif Karagöz, Conceptualization, Resources, Formal analysis, Validation, Investigation, Visualization, Methodology, Writing—original draft, Project administration, Writing—review and editing; Diego Acosta-Alvear, Resources, Methodology, Writing—review and editing, Generated mouse embryonic fibroblast cell line stably expressing wild type hIRE1a-GFP; Hieu T Nguyen, Formal analysis, Validation, Investigation, Ran the XL-MS experiments and analyzed the data; Crystal P Lee, Resources, Methodology, Helped generating mouse embryonic fibroblast cell line stably expressing wild type hIRE1a-GFP; Feixia Chu, Formal analysis, Supervision, Funding acquisition, Validation, Investigation, Methodology, Writing;review and editing, Helped the experimental design of cross-linking coupled to mass spectroscopy experiments (XL-MS), Ran the XL-MS experiments and analyzed the data; Peter Walter, Formal analysis, Supervision, Funding acquisition, Validation, Investigation, Methodology, Writing—review and editing; Conceptualization, Supervision, Funding acquisition, Project administration, Writing—review and editing

## Author ORCIDs

G Elif Karagöz (iD) http://orcid.org/0000-0002-3392-2250

Peter Walter (iD) https://orcid.org/0000-0002-6849-708X

## Decision letter and Author response

Decision letter https://doi.org/10.7554/eLife.30700.033

Author response https://doi.org/10.7554/eLife.30700.034

## Additional files

### Supplementary files

• Transparent reporting form

DOI: https://doi.org/10.7554/eLife.30700.032

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
