## [Decision Letter]

[Editors’ note: a previous version of this study was rejected after peer review, but the authors submitted for reconsideration. The first decision letter after peer review is shown below.]

Thank you for submitting your work entitled "An unfolded protein-induced conformational switch activates mammalian IRE1" for consideration by *eLife*. Your article has been evaluated by a Senior Editor and four reviewers, one of whom is a member of our Board of Reviewing Editors.

Our decision has been reached after consultation between the reviewers. Based on these discussions and the individual reviews below, we regret to inform you that your work will not be considered further for publication in *eLife*.

As noted by the reviewers (attached below), although it is clear that IRE cLD can interact with peptides and unfolded proteins, it is not clear if this is physiologically relevant. The experimental evidence of structural changes on the basis of the NMR experiments are weak. Since it is important to ascertain that the mechanism of IRE1 activation in eukaryotes is conserved, other well established biophysical techniques should be considered to obtain more compelling and conclusive data. Based on the data presented, it is not possible to conclude that the mechanism of substrate binding induces activation of IRE1. We would, however, encourage you to consider submission of a new paper when you are able to address the concerns raised by the reviewers.

Reviewer #1:

The manuscript by Karagoz et al. reports a study on the mammalian homolog of the ER stress sensor IRE1. IRE1 is a membrane protein conserved from yeast to humans, which signals protein folding stress in the ER lumen to the nucleus. This occurs by a cascade of events, with oligomerization of IRE1 as the first crucial step. Displacement of the ER-resident Hsp70 molecular chaperone (BiP) from IRE1 was implicated in oligomerization. However, the group of Peter Walter also convincingly showed binding of misfolded proteins can directly activate yeast IRE1. The crystal structure of the ER-lumenal domain of yeast IRE1 (yIRE1 cLD) exhibited a hydrophobic groove across a dimer interface and yIRE1 cLD was shown to bind a peptide of the misfolding mutant protein carboxypeptidase Y (CPY*). Structural data for such an interaction is however missing. Furthermore, the crystal structure of human IRE1 exhibited a cleft too narrow for a peptide substrate, among other structural differences, challenging the model that hIRE1 uses a similar activation mechanism as yIRE1. The present study uses peptide arrays and NMR to demonstrate that the mechanism of IRE1 oligomerization and hence activation by unfolded protein is conserved in hIRE1.

The presented data on the IRE1 substrate selectivity appears solid. The experimental evidence for structural changes in IRE1 are consistent with but are not conclusive for the proposed model of substrate binding induced activation of IRE1. Since it is important to ascertain that the mechanism of IRE1 activation in eukaryotes is conserved, more direct evidence is required, for example by introducing FRET probes into the helices lining the proposed binding cleft to monitor the closed to open transition as proposed in the model in Figure 7. Hydrogen-deuterium exchange experiments could also provide a more comprehensive picture. Another approach would be to screen for a peptide that binds stably to the luminal domain of IRE1, and then to determine the crystal structure of the complex. This approach has been successful for molecular chaperones such as Hsp70 and GroEL.

Major points:

1) From their peptide array analyses, the authors deduce a preference of hIRE1 for hydrophobic peptides enriched in tyrosine, tryptophan and arginine, similar to yeast IRE1. The minimum peptide length appears to be 12 residues. The dissociation constants for two model peptides (MPZ1 and 8ab1) are rather high (24 and 5 μM, respectively). A peptide of 12 residues binding in an extended conformation in the 40 Å long groove of yeast IRE1 is expected to have a considerably higher affinity, given the extensive surface area that would become buried – the more limited Hsp70-peptide interaction with the NRLLLTG peptide is two orders of magnitude tighter. Surprisingly, the dissociation constants for two model substrate proteins are in the same range. It thus remains unclear how IRE1 saturation can be reached under physiological conditions, where concentrations are presumably lower.

2) To support and validate the proposed mechanism for substrate binding and hIRE1 oligomerization, the authors use NMR spectroscopy of derivatives of the hIRE1 cLD in which the γ-methyl groups of isoleucine are 13C-labeled. Unfortunately, the NMR signals of 6 out of 12 isoleucine methyl groups overlap and cannot be assigned. The mutant L186I provides an additional probe. Still, with only six probe positions, the sampling for the 40 kDa large protein is quite sparse. All NMR signals exhibit broad line width, consistent with chemical exchange on the NMR time scale resulting from hIRE1 cLD forming dimers and tetramers. Binding of the model peptides (MPZ1 or 8ab1) and the model unfolded substrate (CH1) slightly shifts the NMR signals of a subset of the methyl groups. Substrate binding also results in the formation of higher oligomeric states of hIRE1 cLD, suggesting that the shifts observed in the NMR peaks might not only be due to substrate binding but also as a result of oligomerization (conformational changes). The suppression of signals of methyl groups close to the putative peptide-binding groove by a nitroxide spin label in MPZ1 is interpreted as indication that MPZ1-proxyl binds in the center of the groove in hIRE1 cLD. Since optimal range by the PRE effect is 10-25 Å, NOE signals, which have a much narrower range to 5 Å distance, between a labeled peptide and the IRE1 methyl sites could prove the postulated binding mode beyond a reasonable doubt. Photo-crosslinking combined with mass spectrometry could be an alternative approach. Although, mutations at the bottom of the putative binding cleft increased the peptide dissociation constant, the effect was relatively minor (factor of two). Again, this could be due to indirect effects of conformational changes in hIRE1.

Other points:

1) Subsection “Peptide recognition by hIRE1 cLD relies on specific interactions”, first paragraph. The amino acid sequences of the peptides MPZ1 and 8ab1 should be stated when first mentioned. Figure 2 shows the sequence of MPZ1 but the sequence of the peptide 8ab1 is not stated anywhere in the manuscript.

2) Figure 1. There seems to be some divergence in the binding intensities for hIRE1 and yIRE1 in the middle part of CPY*. hIRE1 seems to be more sticky than yIRE1.

3) Legend Figure 1. I assume the peptide sequences with binding scores in the upper 10% segment were selected for the amino acid frequency analysis. This is not clear in the text.

4) In Figure 4, the dimeric peak did not change in the absence or presence of a peptide, while the tetrameric peak declined with the concomitant appearance of larger oligomers. Is it possible that tetramers and not dimers, bind peptides? What is the effect on the oligomeric state of 5 μM hIRE1 cLD in the presence of peptide? For example, does 8ab1 peptide (KD of 5 μM) induce more oligomerization in the AUC assay than MPZ1 peptide (KD of 24 μM)?

5) In Figure 6, it is shown that mutations in the groove decrease the binding affinity of peptides. AUC analysis of the mutants (5 and 75 μM) in the absence and presence of the peptides MPZ1 and 8ab1 would be important to exclude that the mutations have not in any way altered the characteristics of hIRE1 cLD. Furthermore, this analysis will demonstrate that the oligomerization of these mutants in presence of MPZ1-N peptide is inefficient, consistent with the low binding affinity of mutant proteins to MPZ1-N peptide.

Reviewer #2:

This is a very interesting paper that uses both biochemical and biophysical approaches to address mechanistic aspects of function of IRE1 activation. One model suggests that binding of unfolded polypeptides leads to oligomerization that in turn activates function while a second model suggests that other chaperones are released from IREI that leads to regulation. The authors establish conclusively that unfolded molecules are substrates for mammalian IREI and go on to further characterize the substrate sequence preferences. They then show that while there is some sequence preference overlap for BIP and IREI there are also differences that are important given the concentration differences of these two molecules in the cell.

1) Can the authors refute the BiP binding model by showing that it does not interact with IRE1 by NMR or is there known to be an interaction? Given that they have suitably labeled samples this seems like an easy experiment.

2) If the cLDs of human and yeast are functionally interchangeable does this not lend support to the fact that there is a similar mechanism for both human and yeast forms, even without doing any experiments, although I agree that the experiments done do support this much more strongly?

3) Interestingly the authors show that unfolded proteins binding with similar affinity as peptides. I might have assumed there to be a higher affinity for the proteins. If similar microscopic binding affinities for different sites are found then increasing the number of these sites should increase affinity by avidity. I would assume that protein substrates would have more potential binding sites.

I am concerned about some of the conclusions that the authors make on the basis of NMR data. The data is rather marginal and the changes upon binding are small.

4) I could not find whether the authors used 2H protein. While more expensive to produce this will significantly influence the quality of spectra. A molecular weight of 80 kDa is not very large for the methyl-TROSY experiments and yet the spectra look relatively poor.

5) Given that at NMR concentrations there appears to be an equilibrium between different populations might the authors wish to work at higher concentrations so as to push the equilibrium to a more stable state? Would this increase the affinity for substrate, as the active state is "preformed" (because the shifts upon substrate binding seem to mirror those upon increasing the protein concentration)? Given the similarities in the shifts in both cases, can the authors speculate where the tetramerization interface is (is it consistent with the IF2^L^ site). Would one not expect difference in the shifts between concentration and binding as well due to local effects?

6) How do the authors distinguish between stabilization of a discrete higher molecular weight species (i.e., equilibrium shift perhaps to a tetramer) from minimizing excursions between different states within the dimer structure of the cLD. Could the authors perform 2H spin relaxation experiments that are sensitive to size but not chemical exchange to show an oligomerization process. Additionally from the AUC trace it looks like large oligomers are formed (not just tetramers) upon binding unfolded substrate. So I might have thought that the NMR spectrum would get worse. Please clarify. In this context what about making mutations at the putative IF2 site and showing that the spectra get much better and that affinity for peptide is lower (presumably since it is not accompanied by a shift in equilibrium that supports binding). This would be a powerful experiment to show that indeed there is a dynamic equilibrium between oligomer states.

7) The authors write: "Interestingly, the isoleucine peaks corresponding to Ile52 and Ile263 shifted in the same direction and to a similar extent when hIRE1 cLD bound to MPZ1-N peptide (Supplementary Figure 6)."

Unfortunately the shifts are very tiny and hard to quantify. For I52 it looks like one sees peaks from 2 states (dimer, tetramer) in the 25 μm IRE1 cLD sample, yet for I263 there is only 1 peak in the free form? For I264 it is very hard to see that the shifts are similar upon increasing concentration and upon binding, again reflecting such small shifts in the first place. To conclude that the structures of the "oligomers" are similar in both cases from such small CSPs and from such few probes seems a stretch to me.

8) The authors write: "Ile128 and Ile52 peaks were slightly affected by the binding of MPZ1-proxyl peptide, supporting the idea that these isoleucines rearrange upon peptide binding".

Why? They can simply be further away from the spin label and show smaller effects. Given the 1/r^6 nature of the PRE small changes in distances can have rather large changes in intensities.

9) The authors use mutagenesis to confirm the importance of certain residues in the binding. Although mutations do change affinities by 2 fold or so (16 μm to 32 μm for example) this is very little difference in DG. One might have expected more. Does this mean that there is a highly dynamic complex with contributions from many sites? Might some insight be obtained by doing reverse experiments whereby the amide spectrum of the substrate is recorded to see if there are dynamics visible from this end? It is noteworthy that single point mutants in the substrate can impair binding yet corresponding point mutants in the cLD seem to have a small effect on free energy differences.

10) A concluding point is "Second, NMR spectroscopy revealed that peptides bind to hIRE1 cLD's MHC446 like groove and induce a conformational change including the distant αB helix,…"

Not sure I see this. Seems like the data is not of sufficient quality to support this. How many probes are in helix B (I could find only one in the text).

11) "Taken together, our data converge on a model, which poses that unfolded protein-binding activates a switch in IRE1's cLD, stabilizing its active conformation, which is prone to oligomerization after rearrangements that render it compatible with the formation of IL2L"

Yet what are the rearrangements?

Reviewer #3:

In this manuscript, the authors characterized the direct binding of unfolded protein model substrates to the MHC-like groove in human IRE1 ER-lumenal domain and the conformational switch occurring within the domain. As there is an outstanding debate on IRE1 sensing mechanism from ER stress, the authors compared the binding preferences of IRE1 core-lumenal domain (IRE1 cLD) and BiP to a peptide array derived from different protein substrates. IRE1 cLD and BiP exhibited distinctive binding preferences. Next, they demonstrated that IRE1 cLD can bind peptides derived from protein substrates with an affinity similar to chaperone-unfolded protein interaction. These indicate that IRE1 cLD is capable of directly binding unfolded proteins and thus suggesting it might directly sense the accumulation of unfolded proteins in vivo. To further characterize IRE1 sensing mechanism from unfolded proteins, the authors employed advanced NMR techniques to determine substrate binding and conformation changes that occur within IRE1 cLD by labeling specific residues detectable by NMR. They also used size exclusion chromatography and analytical ultracentrifugation sedimentation velocity to quantify IRE1 cLD oligomeric states. From these experiments, the authors concluded that ( et al., 2010) IRE1 cLD forms larger oligomers in a concentration dependent manner as well as in the presence of substrates; (2) substrate binding induces IRE1 cLD conformation changes; (3) the substrate binding site is localized to the centre of IRE1 MHC-like groove.

The major advance for the fields, from their findings, is the characterization of substrate binding to IRE1 cLD. This will add very strong weight on the model where IRE1 can directly senses the accumulation of unfolded protein instead of being activated by the detachment of BiP through IRE1 lumenal domain. They have accumulated extensive and rigorous in vitro data to support their model. Overall, their experiments are clean and very well thought. Their study will definitely be of great interest in the ER stress field and beyond. I highly recommend the manuscript to be published in *eLife*. However, there are few points that should be clarified by the authors.

1) They identified and used, throughout the study, a specific short peptide of mouse MPZ protein (MPZ1-N) which strongly binds hIRE1 cLD. MPZ1-N is part of the cytosolic domain of ER stress model substrate MPZ suggesting hIRE1 will never bind MPZ1-N in vivo. Similarly, BiP seems to bind only the cytosolic domain of MPZ from Figure 1. These peptides can be seen as model substrates but the authors should address this discrepancy not to give the impression that the UPR is activated through IRE1 by recognizing the cytosolic domain of unfolded MPZ as suggested in the subsection “The lumenal domain of human IRE1 binds peptides”.

2) hIRE1 cLD is proposed to form large oligomers in the presence of MPZ1-N but the authors don't address the issue of how the dimers associate to form higher order oligomers in vitro.

Reviewer #4:

The manuscript by Karagoz, Chen and Walter explores the hypothesis that human IRE1 responds to ER stress through direct binding of unfolded substrate proteins, as has been proposed for yeast IRE1. Differences in the crystal forms of these two proteins brought the generality of the hypothesis proposed based on the presence of a possible substrate binding cleft in the yeast protein, but absent in the human, into question. The authors have some compelling data, but these are mixed with some less compelling observations (at best), and some findings that are very complicated to interpret, in this reviewer's estimation. It would seem advisable to present the data that can be straightforwardly interpreted and are necessary to make the authors' case (i.e., that hIRE1 binds substrates), and to remove some of the weaker and more confusing experiments. In addition to the complexity of the system, there are several examples of sloppiness in the data presentation that detract from this potentially interesting and provocative paper. These must be rectified. As summarized in the points raised below, the uneven quality of the data in this study and the complexity of the proposed equilibria weaken this paper; the structural hypothesis is attractive and it would be nice to have a more compelling test of it.

1) The binding of peptides in arrays prepared from several potential substrates is a straightforward and well analyzed result. It is provocative and physiologically interesting that there is a preference for sequences and residues that is similar but not identical to that of BiP. It is puzzling, however, that the authors didn't compare the yeast and human IRE1 proteins on the same arrays.

2) The authors present binding curves for many of the peptides and confirm that a couple of proteins bind as well. They report Kd's. However, in the next breath they show that the human IRE1 luminal domain equilibrates between a dimer and a tetramer and upon peptide binding shifts to higher order oligomers. What then is the meaning of the binding data? How do they know which species the peptide is binding? How is the apparent affinity interpreted when an allosteric quaternary structure shift accompanies the binding? In my opinion, they should show the substrate-dependent oligomerization and develop their hypothesis for its origin and its importance, and skip the quantitative binding analysis, which is likely not valid. [Note: Panel A of Figure 3 does not belong in that figure, as it shows binding of a protein substrate, and nothing to do with oligomerization, the theme of the figure.]

3) In Figure 3, there are insufficient labels on panel D, and the concentration is not given for the NMR spectrum. Would it not be informative to run the NMR spectrum at different concentrations to see the shifts that are associated with oligomerization? The broadening is really a challenging aspect of the analysis of the NMR data.

4) In Figure 4, the Ile methyl data are very confusing. The spectra in black are supposed to be the apo-protein, but they are not the same for the columns for two different substrates. Why not? And the chemical shifts are confusing. In one case (Ile 128), they go down from the top of the y axis to the bottom, but in all other cases they go up. Also, the labeling of the axes should be done in a more reasonable way with the same labeling across (the same Ile) or up (the same peptide).

5) Figure 4, panel D is also confusing because the amount of dimer seems not to be changing, but the tetramer seems to go towards higher order oligomers. This is not the scheme presented in the authors' model.

6) The chemical shift perturbation data clearly show that Ile128 experiences a much larger shift than the others, but the coloring on the structure suggests anything above the threshold is equally perturbed. More gradations are needed. More importantly, how are shifts from the associated with oligomerization distinguished from those due to binding? The more compelling data come from the spin labeled peptide, which clearly shows binding to occur near Ile186. However, the weak signals from several Ile peaks make these data a bit sketchy. Certainly some error bars should be given. [Where is the other Ile128 peak?]

7) I am not persuaded by the changes in affinity due to mutation, given the complexity of the equilibria going on – both oligomerization and binding – and the small changes in the binding curves.

[Editors’ note: what now follows is the decision letter after the authors submitted for further consideration.]

Thank you for submitting your article "An unfolded protein-induced conformational switch activates mammalian IRE1" for consideration by *eLife*. Your article has been favorably evaluated by Randy Schekman (Senior Editor) and four reviewers, one of whom is a member of our Board of Reviewing Editors. The reviewers have opted to remain anonymous.

The reviewers have discussed the reviews with one another and the Reviewing Editor has drafted this decision to help you prepare a revised submission.

The consensus was that this revision takes into account a majority of the concerns associated with the initial submission, although there are a number of outstanding issues that we would like addressed in a further revision. Additional experiments are not required. It was felt that there is some degree of over-interpretation of the NMR data, as the perturbations to chemical shifts were extremely small. Additional discussion as to this limitation should be given in a revised manuscript. Further, more discussion about the role of the BiP chaperone should be provided, including a discussion about the possibility of a model which involves both BiP and unfolded proteins playing a role in activating IRE1.

1) A major concern remains with the interpretation of some of the NMR data. The shifts are small and this places certain limits on what can be said. The data do support the notion of peptide binding and are consistent with binding occurring in the MHC-like groove. Yet the authors use a PRE approach to help establish a conformational change that accompanies binding, suggesting that such a change occurs and is allosteric in nature. With the level of s/n present in these experiments it is clear that 186 is close to the label and perhaps 124 (although s/n is really poor here). In the supplementary figures it even seems like the spin-label spectra has cross-peaks with more intensity but this must be due to the fact that the spectra are not plotted properly (see Figure 5—figure supplement 1 for peaks other than 186). It is not clear how one can conclude from this that there are conformational changes. It seems that the authors should confront the problem head-on by saying that while the NMR data does not allow one to establish conformational changes to the structure, the observation of the large PRE to 186 supports the notion of binding to the MHC groove which is only possible if there is a change in conformation. A couple of more points about this topic:

A) Is there a mistake in Figure 3? Is the assignment of 124 correct or do you mean 128?

B) I wonder if your approach to normalize the PREs by surface accessibility is a wise one. Of course residues at the surface can have larger PREs than a proximal but buried residue. But the net result is that in your Figure 5 there is little dynamic range. It is not immediately obvious how to interpret these numbers – a value of 0.13 for 124 is much more significant than 0.2 for 52?

C) In the subsection “Oligomerization leads to global conformational changes in hIRE1α cLD”, second paragraph, it speaks about the chemical shift changes to 124 and 128 as indicative of conformational changes. This seems questionable. In the first paragraph of the Discussion, it speaks about binding of peptides inducing conformational changes to distant helices. Again, the changes are tiny.

2) It seems that the field of ER stress seeks to have one or the other model of regulation of the unfolded protein response win out. Why is it not possible that Ire1 responds both to BiP concentration and unfolded protein load?

3) The binding studies conducted by the authors involved peptides. Can the authors comment on binding of intact proteins as well?

4) A number of concerns were raised concerning the identification of human Ire1 binding sequences. There were no sequences in the MPZ lumenal domain that interacted with recombinant Ire1α-cLD, whereas the sequence that was used for the bulk of experiments is located in the cytosolic domain. Although this doesn't interfere with the conclusion that human Ire can bind peptides, the conclusions about the type of sequences that it can bind may not be reliable, since this protein does induce a UPR in mammalian cells. A second point involves the Sars-CoV, 8ab protein. The paper cited for this protein demonstrates that it binds to ATF6 and induces its cleavage but has no effect on Ire1 or PERK activation. So if a peptide from this protein can bind to Ire1, it is not sufficient to activate it. Lastly, the CH1 domain produced by Marcinowski et al. was oxidized, which resulted in BiP binding to it with BiP's lid in a much more open conformation than when peptides bound to it. Do the authors find that the perturbation of Ire1 luminal domain is much greater in order to accommodate this client than when peptides are used in the NMR studies?

---

## [Author Response]

[Editors’ note: the author responses to the first round of peer review follow.]

Reviewer #1:Major points:1) From their peptide array analyses, the authors deduce a preference of hIRE1 for hydrophobic peptides enriched in tyrosine, tryptophan and arginine, similar to yeast IRE1. The minimum peptide length appears to be 12 residues. The dissociation constants for two model peptides (MPZ1 and 8ab1) are rather high (24 and 5 μM, respectively). A peptide of 12 residues binding in an extended conformation in the 40 Å long groove of yeast IRE1 is expected to have a considerably higher affinity, given the extensive surface area that would become buried – the more limited Hsp70-peptide interaction with the NRLLLTG peptide is two orders of magnitude tighter. Surprisingly, the dissociation constants for two model substrate proteins are in the same range. It thus remains unclear how IRE1 saturation can be reached under physiological conditions, where concentrations are presumably lower.

The reviewer addressed an important point. We now clarified the issue in the Discussion. It is clearly possible that not all amino acids of the peptide form contacts with the groove. Instead it seems more plausible that certain side chains act as anchor residues similar to peptide binding to MHC molecules. Unlike ATP regulated Hsp70s, IRE1 LD’s interaction with unfolded proteins is intrinsically dynamic. Importantly, we now show that MPZ1-N-2X peptide that harbors two IRE1 binding sites, binds tightly to hIRE1 cLD due to increased avidity (Figure 2). Moreover, we anticipate that when protein folding homeostasis is disrupted in the ER, the critical concentration for this interaction would be met at the ER-membrane where the co-translational folding events take place.

2) To support and validate the proposed mechanism for substrate binding and hIRE1 oligomerization, the authors use NMR spectroscopy of derivatives of the hIRE1 cLD in which the γ-methyl groups of isoleucine are 13C-labeled. Unfortunately, the NMR signals of 6 out of 12 isoleucine methyl groups overlap and cannot be assigned. The mutant L186I provides an additional probe. Still, with only six probe positions, the sampling for the 40 kDa large protein is quite sparse. All NMR signals exhibit broad line width, consistent with chemical exchange on the NMR time scale resulting from hIRE1 cLD forming dimers and tetramers. Binding of the model peptides (MPZ1 or 8ab1) and the model unfolded substrate (CH1) slightly shifts the NMR signals of a subset of the methyl groups. Substrate binding also results in the formation of higher oligomeric states of hIRE1 cLD, suggesting that the shifts observed in the NMR peaks might not only be due to substrate binding but also as a result of oligomerization (conformational changes). The suppression of signals of methyl groups close to the putative peptide-binding groove by a nitroxide spin label in MPZ1 is interpreted as indication that MPZ1-proxyl binds in the center of the groove in hIRE1 cLD. Since optimal range by the PRE effect is 10-25 Å, NOE signals, which have a much narrower range to 5 Å distance, between a labeled peptide and the IRE1 methyl sites could prove the postulated binding mode beyond a reasonable doubt. Photo-crosslinking combined with mass spectrometry could be an alternative approach.

We now impaired oligomerization of hIRE1 cLD via mutations and mapped peptide binding to this mutant by NMR spectroscopy. Peptide binding to this mutant resulted in more significant shifts in isoleucine peaks in the groove (see Figure 7).

Due to its oligomerization propensity, hIRE1 cLD is a challenging target for NMR studies. The peaks are low intensity due to the line broadening caused by the chemical exchange. These properties make hIRE1 cLD inaccessible to NOE type of experiments. Yet, even though we have limited number of probes in the NMR measurements, this non-invasive method provides very precise information about the conformational changes in the molecule at atomic resolution. We now could follow the conformational changes in this large and heterogeneous molecule upon peptide binding and oligomerization. The isoleucines are mostly buried in the protein so that they are not as sensitive to binding events. We and others have seen similar phenomena in different systems using this approach (Karagoz et al., PNAS, 2011 and Cell 2014, Rosenzweig et al., Science 2013, Stoffregen et al., Structure, 2012).

Supporting this notion, we see larger shifts of threonine peaks upon peptide binding (see Figure 4). Even if small, the effects of binding of various peptides and proteins on the isoleucine peaks are highly reproducible and entirely consistent between experiments.

Although, mutations at the bottom of the putative binding cleft increased the peptide dissociation constant, the effect was relatively minor (factor of two). Again, this could be due to indirect effects of conformational changes in hIRE1.

We agree with the reviewer’s point. We now omitted these data from the manuscript. Unfortunately mutating the aromatic residues in the groove, which we anticipate to play an important role for hIRE1 cLD’s interaction with unfolded proteins, results in destabilization of the protein. We were not able to obtain data from these mutants. As the reviewer mentions, there is only two-fold drop in the affinity due to a single point mutation at Thr159. However, this is expected as this residue most likely forms a hydrogen bond with the peptide.

Other points:1) Subsection “Peptide recognition by hIRE1 cLD relies on specific interactions”, first paragraph. The amino acid sequences of the peptides MPZ1 and 8ab1 should be stated when first mentioned. Figure 2 shows the sequence of MPZ1 but the sequence of the peptide 8ab1 is not stated anywhere in the manuscript.

We corrected this omission in the text.

2) Figure 1. There seems to be some divergence in the binding intensities for hIRE1 and yIRE1 in the middle part of CPY*. hIRE1 seems to be more sticky than yIRE1.

The reviewer is right. We now removed these data from the manuscript to improve the flow. If reviewers find it necessary, we would be happy to insert the data.

3) Legend Figure 1. I assume the peptide sequences with binding scores in the upper 10% segment were selected for the amino acid frequency analysis. This is not clear in the text.

We clarified the text accordingly (subsection “The lumenal domain of human IRE1α binds unfolded polypeptides”).

4) In Figure 4, the dimeric peak did not change in the absence or presence of a peptide, while the tetrameric peak declined with the concomitant appearance of larger oligomers. Is it possible that tetramers and not dimers, bind peptides? What is the effect on the oligomeric state of 5 μM hIRE1 cLD in the presence of peptide? For example, does 8ab1 peptide (KD of 5 μM) induce more oligomerization in the AUC assay than MPZ1 peptide (KD of 24 μM)?

We now identified a mutation that impaired the oligomerization of hIRE1α cLD (hIRE1α cLD IF2^L^ mutant), and this mutant binds to peptides with a very similar affinity as to wild type hIRE1α cLD (see Figure 7, Figure 7—figure supplement 3). These data suggest that hIRE1 cLD oligomers do not have higher affinity for peptides.

5) In Figure 6, it is shown that mutations in the groove decrease the binding affinity of peptides. AUC analysis of the mutants (5 and 75 μM) in the absence and presence of the peptides MPZ1 and 8ab1 would be important to exclude that the mutations have not in any way altered the characteristics of hIRE1 cLD. Furthermore, this analysis will demonstrate that the oligomerization of these mutants in presence of MPZ1-N peptide is inefficient, consistent with the low binding affinity of mutant proteins to MPZ1-N peptide.

As already mentioned above, we now omitted the experiments on the groove mutants as the mutation in Threonine159 to alanine only dropped the binding affinity by twofold. Unfortunately, mutating the aromatic residues in the groove resulted in destabilization of the protein.

Reviewer #2:[…] 1) Can the authors refute the BiP binding model by showing that it does not interact with IRE1 by NMR or is there known to be an interaction? Given that they have suitably labeled samples this seems like an easy experiment.

We see an interaction of hIRE1 cLD with BiP in our hands. A characterization of hIRE1 cLD’s interaction with BiP will be the subject of an extensive follow-up manuscript. Inclusion of the data in this manuscript would, in our view, render the paper unfocussed.

2) If the cLDs of human and yeast are functionally interchangeable does this not lend support to the fact that there is a similar mechanism for both human and yeast forms, even without doing any experiments, although I agree that the experiments done do support this much more strongly?

We removed these data from the paper to improve the flow of presentation. However, we would of course be happy to provide the data if the reviewers would find it useful.

3) Interestingly the authors show that unfolded proteins binding with similar affinity as peptides. I might have assumed there to be a higher affinity for the proteins. If similar microscopic binding affinities for different sites are found then increasing the number of these sites should increase affinity by avidity. I would assume that protein substrates would have more potential binding sites.

This is an interesting point. As mentioned above, to test this possibility, we now designed a peptide by repeating the MPZ1-N sequence twice in a peptide sequence (called this peptide MPZ1-N-2X). We showed that MPZ1-N-2X binds tighter to hIRE1 cLD due to avidity (see Figure 2).

I am concerned about some of the conclusions that the authors make on the basis of NMR data. The data is rather marginal and the changes upon binding are small.

The shifts of the isoleucine peaks are small mainly due to two properties: i) they are buried in the structure and ii) hIRE1 cLD exists as a conformational equilibrium in solution. As we can experimentally control only the latter, we now included experiments where we either uncoupled the oligomerization from peptide binding (Figure 7) or compared the concentrations of hIRE1 cLD where it only formed dimers with high concentrations where large fraction of it formed oligomers (Figure 6). In both conditions, we observed larger shifts in isoleucine peaks.

4) I could not find whether the authors used 2H protein. While more expensive to produce this will significantly influence the quality of spectra. A molecular weight of 80 kDa is not very large for the methyl-TROSY experiments and yet the spectra look relatively poor.

We cited an earlier paper in the Materials and methods, where the use of D_2_O was clearly described.

To make this information more accessible, we now added this information to the Materials and methods. We used D_2_O and deuterated glucose during expression to make a highly deuterated sample. Unfortunately, this did not yield high spectral quality because the line broadening is mainly due to chemical exchange in the intermediate exchange regime, not due to relaxation.

5) Given that at NMR concentrations there appears to be an equilibrium between different populations might the authors wish to work at higher concentrations so as to push the equilibrium to a more stable state? Would this increase the affinity for substrate, as the active state is "preformed" (because the shifts upon substrate binding seem to mirror those upon increasing the protein concentration)?

Increasing the concentration of the protein results in poorer spectral quality, due to higher conformational heterogeneity (see Figure 6—figure supplement 2). The hIRE1 cLD mutant, which we impaired oligomerization by mutations (called IF2^L^ mutant) showed similar peptide binding curves as wild type protein, therefore suggesting that tetramers are not better binders. This is addressed in the text (subsection “The peptide-induced allosteric switch remains intact in IF2^L^ mutant hIRE1α cLD”, first paragraph) and in Figure 7 and Figure 7—figure supplement 3.

Given the similarities in the shifts in both cases, can the authors speculate where the tetramerization interface is (is it consistent with the IF2^L^ site).

We now employed crosslinking coupled to mass spectrometry to map the oligomerization interface in hIRE1 cLD and included a paragraph describing the tetramerization interface based on the NMR and crosslinking data (subsection “Identifying the oligomerization interface of hIRE1α cLD”, first paragraph).

Would one not expect difference in the shifts between concentration and binding as well due to local effects?

That is an important point. Definitely peptide binding both modulates the equilibrium between dimers and oligomers and changes hIRE1 cLD conformation. Our data suggest that Ile52 and Ile263 are far from the peptide binding site and that is why the peptide binding induced changes which lead to hIRE1 cLD oligomerization are very similar to the concentration-dependent oligomerization events.

6) How do the authors distinguish between stabilization of a discrete higher molecular weight species (i.e., equilibrium shift perhaps to a tetramer) from minimizing excursions between different states within the dimer structure of the cLD. Could the authors perform 2H spin relaxation experiments that are sensitive to size but not chemical exchange to show an oligomerization process. Additionally from the AUC trace it looks like large oligomers are formed (not just tetramers) upon binding unfolded substrate. So I might have thought that the NMR spectrum would get worse. Please clarify.

The AUC experiments that are performed at NMR concentrations address the very same question. Methyl-TROSY experiments are optimized for much larger proteins. Therefore, the spectrum does not get worse upon formation of larger oligomers. The poor spectral quality of hIRE1 cLD is not due to relaxation but the conformational heterogeneity intrinsic to the biology of the protein. We see improvement of the spectrum when hIRE1cLD binds to peptides, because the binding stabilizes distinct states of hIRE1cLD oligomers.

In this context what about making mutations at the putative IF2 site and showing that the spectra get much better and that affinity for peptide is lower (presumably since it is not accompanied by a shift in equilibrium that supports binding). This would be a powerful experiment to show that indeed there is a dynamic equilibrium between oligomer states.

The hIRE1cLD IF2^L^ mutant, where we impaired hIRE1 cLD oligomerization improved the spectral quality (Figure 7—figure supplement 4), however most isoleucine peaks are still very broad suggesting that the exchange between monomer-dimer states is also very prominent, which is strongly supported by the AUC and gel filtration experiments (see Figure 6 and Figure Supp. 6b).

7) The authors write: "Interestingly, the isoleucine peaks corresponding to Ile52 and Ile263 shifted in the same direction and to a similar extent when hIRE1 cLD bound to MPZ1-N peptide (Supplementary Figure 6)."Unfortunately the shifts are very tiny and hard to quantify. For I52 it looks like one sees peaks from 2 states (dimer, tetramer) in the 25 μm IRE1 cLD sample, yet for I263 there is only 1 peak in the free form? For I264 it is very hard to see that the shifts are similar upon increasing concentration and upon binding, again reflecting such small shifts in the first place. To conclude that the structures of the "oligomers" are similar in both cases from such small CSPs and from such few probes seems a stretch to me.

We agree the shifts are tiny due to conformational heterogeneity. We now added an experiment comparing mainly dimeric hIRE1 cLD (5 µM) with 50 µM and the shifts are much more prominent (see Figure 6).

8) The authors write: "Ile128 and Ile52 peaks were slightly affected by the binding of MPZ1-proxyl peptide, supporting the idea that these isoleucines rearrange upon peptide binding".Why? They can simply be further away from the spin label and show smaller effects. Given the 1/r^6 nature of the PRE small changes in distances can have rather large changes in intensities.We completely agree with the reviewer; we did not state this point clearly in the paper. The point we wanted to make is exactly what the reviewer suggests. We made the change in the text accordingly (see subsection “Peptide binding maps to the MHC-like groove in hIRE1α cLD”).9) The authors use mutagenesis to confirm the importance of certain residues in the binding. Although mutations do change affinities by 2 fold or so (16 μm to 32 μm for example) this is very little difference in DG. One might have expected more. Does this mean that there is a highly dynamic complex with contributions from many sites?

We agree with the reviewer’s point and omitted these data. Please see above in response to reviewer #1, point 2.

Might some insight be obtained by doing reverse experiments whereby the amide spectrum of the substrate is recorded to see if there are dynamics visible from this end? It is noteworthy that single point mutants in the substrate can impair binding yet corresponding point mutants in the cLD seem to have a small effect on free energy differences.

Compared to large protein surface, short peptides can provide limited possible compensatory contacts when a crucial amino acid for binding is mutated. The experiments that monitor the backbone amide groups are not suitable for studying large complexes, as they result in signal loss due to relaxation. We instead monitored the methyl groups of aliphatic side chains of the peptide and saw that certain methyl side chains are more stably engaged with hIRE1 cLD compared to others (data not shown, but we can provide upon request).

10) A concluding point is "Second, NMR spectroscopy revealed that peptides bind to hIRE1 cLD's MHC446 like groove and induce a conformational change including the distant αB helix,…"Not sure I see this. Seems like the data is not of sufficient quality to support this. How many probes are in helix B (I could find only one in the text).

As the reviewer mentioned, we have one probe from the αB helix. In addition, we now obtained data from crosslinking coupled to mass spectrometry experiments. These data showed that Lys265 forms a prominent crosslink site in hIRE1 cLD oligomers, substantiating our NMR data by an independent method (see subsection “Identifying the oligomerization interface of hIRE1α cLD” and Figure 7, Table 1).

11) "Taken together, our data converge on a model, which poses that unfolded protein-binding activates a switch in IRE1's cLD, stabilizing its active conformation, which is prone to oligomerization after rearrangements that render it compatible with the formation of IL2L"Yet what are the rearrangements?

We now described these rearrangements (Discussion, second paragraph).

Reviewer #3:[…] 1) They identified and used, throughout the study, a specific short peptide of mouse MPZ protein (MPZ1-N) which strongly binds hIRE1 cLD. MPZ1-N is part of the cytosolic domain of ER stress model substrate MPZ suggesting hIRE1 will never bind MPZ1-N in vivo. Similarly, BiP seems to bind only the cytosolic domain of MPZ from Figure 1. These peptides can be seen as model substrates but the authors should address this discrepancy not to give the impression that the UPR is activated through IRE1 by recognizing the cytosolic domain of unfolded MPZ as suggested in the subsection “The lumenal domain of human IRE1 binds peptides”.

The reviewer is correct with the topology. We added the topology of the protein and where MPZ1 peptide is found in the protein sequence to Figure 2—figure supplement 1.

2) hIRE1 cLD is proposed to form large oligomers in the presence of MPZ1-N but the authors don't address the issue of how the dimers associate to form higher order oligomers in vitro.

We described the structural arrangements leading to hIRE1 cLD oligomerization suggested by our data in the text, please see Discussion, second paragraph.

Reviewer #4:The manuscript by Karagoz, Chen and Walter explores the hypothesis that human IRE1 responds to ER stress through direct binding of unfolded substrate proteins, as has been proposed for yeast IRE1. Differences in the crystal forms of these two proteins brought the generality of the hypothesis proposed based on the presence of a possible substrate binding cleft in the yeast protein, but absent in the human, into question. The authors have some compelling data, but these are mixed with some less compelling observations (at best), and some findings that are very complicated to interpret, in this reviewer's estimation. It would seem advisable to present the data that can be straightforwardly interpreted and are necessary to make the authors' case (i.e., that hIRE1 binds substrates), and to remove some of the weaker and more confusing experiments. In addition to the complexity of the system, there are several examples of sloppiness in the data presentation that detract from this potentially interesting and provocative paper. These must be rectified. As summarized in the points raised below, the uneven quality of the data in this study and the complexity of the proposed equilibria weaken this paper; the structural hypothesis is attractive and it would be nice to have a more compelling test of it.1) The binding of peptides in arrays prepared from several potential substrates is a straightforward and well analyzed result. It is provocative and physiologically interesting that there is a preference for sequences and residues that is similar but not identical to that of BiP. It is puzzling, however, that the authors didn't compare the yeast and human IRE1 proteins on the same arrays.

We did this experiment for CPY* but removed these data from the manuscript to improve the flow. If reviewers find it necessary, we would be happy to insert the data.

2) The authors present binding curves for many of the peptides and confirm that a couple of proteins bind as well. They report Kd's. However, in the next breath they show that the human IRE1 luminal domain equilibrates between a dimer and a tetramer and upon peptide binding shifts to higher order oligomers. What then is the meaning of the binding data? How do they know which species the peptide is binding? How is the apparent affinity interpreted when an allosteric quaternary structure shift accompanies the binding?

That is an important point, we now addressed this in the text (Results section). We now impaired IRE1 cLD oligomerization by a mutation and the peptide binding curves we obtained with this mutant are very similar to wild type hIRE1 cLD. These data suggested that hIRE1 cLD dimers bind peptides similar to oligomeric species.

In my opinion, they should show the substrate-dependent oligomerization and develop their hypothesis for its origin and its importance, and skip the quantitative binding analysis, which is likely not valid. [Note: Panel A of Figure 3 does not belong in that figure, as it shows binding of a protein substrate, and nothing to do with oligomerization, the theme of the figure.]

We now modified the figures accordingly.

3) In Figure 3, there are insufficient labels on panel D, and the concentration is not given for the NMR spectrum. Would it not be informative to run the NMR spectrum at different concentrations to see the shifts that are associated with oligomerization? The broadening is really a challenging aspect of the analysis of the NMR data.

We added the concentrations of the NMR data in the figure legends. To address reviewers point, we now also compared hIRE1 cLD at 5 µM versus 50 µM concentration (see Figure 6).

4) In Figure 4, the Ile methyl data are very confusing. The spectra in black are supposed to be the apo-protein, but they are not the same for the columns for two different substrates. Why not? And the chemical shifts are confusing. In one case (Ile 128), they go down from the top of the y axis to the bottom, but in all other cases they go up.

This is an important point. The splitting of peaks in different spectra is due to differences in the concentration of the apo protein in different experiments. We now repeated all the experiments with same hIRE1 cLD concentration when we compared the binding of different peptides to exclude the effect of different oligomeric states to the binding events. Please see Figure 4.

Also, the labeling of the axes should be done in a more reasonable way with the same labeling across (the same Ile) or up (the same peptide).

We modified the figures accordingly.

5) Figure 4, panel D is also confusing because the amount of dimer seems not to be changing, but the tetramer seems to go towards higher order oligomers. This is not the scheme presented in the authors' model.

The reviewer is right. We now did AUC experiments at lower hIRE1 cLD concentrations and with hIRE1 cLD IF2^L^ mutant that impaired hIRE1 cLD oligomerization (Figure 6 and Figure 7—figure supplement 3). These experiments showed that peptide binding stabilizes hIRE1 cLD dimers in these conditions. Therefore, our data converge on the model presented in Figure 9.

6) The chemical shift perturbation data clearly show that Ile128 experiences a much larger shift than the others, but the coloring on the structure suggests anything above the threshold is equally perturbed. More gradations are needed. More importantly, how are shifts from the associated with oligomerization distinguished from those due to binding? The more compelling data come from the spin labeled peptide, which clearly shows binding to occur near Ile186. However, the weak signals from several Ile peaks make these data a bit sketchy. Certainly some error bars should be given. [Where is the other Ile128 peak?]

We now identified a mutant that impairs hIRE1 cLD oligomerization (IF2^L^ mutant) and repeated the NMR experiments with this mutant. These experiments show larger shifts of the isoleucine peaks in the groove residues (Figure 7) upon peptide binding. Moreover, the signal-to-noise of the isoleucine peaks in the PRE experiments is quite high and the PRE effects are highly reproducible.

7) I am not persuaded by the changes in affinity due to mutation, given the complexity of the equilibria going on – both oligomerization and binding – and the small changes in the binding curves.

We removed the figure about changes in the affinity in the select groove residues (please see our response to reviewer 1, point 2 above). Moreover, now we show that hIRE1 IF2^L^ mutant, which cannot form oligomers, does not alter hIRE1 cLD’s apparent affinity for peptides. These data suggest that oligomerization does not modulate hIRE1 cLD affinity for peptides (see Figure 7 and Figure 7—figure supplement 3).

[Editors' note: the author responses to the re-review follow.]

The consensus was that this revision takes into account a majority of the concerns associated with the initial submission, although there are a number of outstanding issues that we would like addressed in a further revision. Additional experiments are not required. It was felt that there is some degree of over-interpretation of the NMR data, as the perturbations to chemical shifts were extremely small. Additional discussion as to this limitation should be given in a revised manuscript. Further, more discussion about the role of the BiP chaperone should be provided, including a discussion about the possibility of a model which involves both BiP and unfolded proteins playing a role in activating IRE1.1) A major concern remains with the interpretation of some of the NMR data. The shifts are small and this places certain limits on what can be said. The data do support the notion of peptide binding and are consistent with binding occurring in the MHC-like groove. Yet the authors use a PRE approach to help establish a conformational change that accompanies binding, suggesting that such a change occurs and is allosteric in nature. With the level of s/n present in these experiments it is clear that 186 is close to the label and perhaps 124 (although s/n is really poor here). In the supplementary figures it even seems like the spin-label spectra has cross-peaks with more intensity but this must be due to the fact that the spectra are not plotted properly (see Figure 5—figure supplement 1 for peaks other than 186). It is not clear how one can conclude from this that there are conformational changes. It seems that the authors should confront the problem head-on by saying that while the NMR data does not allow one to establish conformational changes to the structure, the observation of the large PRE to 186 supports the notion of binding to the MHC groove which is only possible if there is a change in conformation. A couple of more points about this topic:

Here, we used chemical shift perturbation analyses to monitor conformational changes in hIRE1 cLD upon peptide binding by using isoleucines and threonines as NMR visible probes. The chemical perturbation experiments show that peptide binding changes the environment of select residues for both isoleucines and threonines for different peptides and a model unfolded protein. To substantiate our findings, we complemented the chemical shift mapping with PRE approach, where we attach a spin label to a peptide to identify residues in the vicinity of the peptide-binding site. The PRE experiments together with the chemical shift perturbation analyses with the wild type IRE1 and IF2 mutant made it possible to define the isoleucines that are in close proximity to the binding site and the ones that are further away and yet shifting due to peptide induced conformational changes. As we described earlier, the changes in isoleucines are not very large as these isoleucines are mostly buried and less sensitive to binding events, we now clarified this point in the manuscript (subsection “Peptide binding stabilizes the open conformation of hIRE1α cLD”). To increase the number of NMR probes and provide additional evidence, we used threonines as NMR markers. Even though these residues are not assigned, threonine peaks display larger shifts upon peptide binding and support a peptide induced conformational change in hIRE1 cLD.

A) Is there a mistake in Figure 3? Is the assignment of 124 correct or do you mean 128?

Yes, this was a typo in the Figure 3. We have corrected the mistake. Thank you!

B) I wonder if your approach to normalize the PREs by surface accessibility is a wise one. Of course residues at the surface can have larger PREs than a proximal but buried residue. But the net result is that in your Figure 5 there is little dynamic range. It is not immediately obvious how to interpret these numbers – a value of 0.13 for 124 is much more significant than 0.2 for 52?

Unlike most of the studies where PRE experiments are used for mapping intact protein-protein interactions, we find the normalization of the surface exposure to be important for the experiments conducted with peptides. To address the reviewer’s concern about the significance of the broadening of Ile124 versus Ile52, we now placed a more stringent cut off in Figure 5, which placed Ile124 in the same category as Ile128 and Ile52.

C) In the subsection “Oligomerization leads to global conformational changes in hIRE1α cLD”, second paragraph, it speaks about the chemical shift changes to 124 and 128 as indicative of conformational changes. This seems questionable. In the first paragraph of the Discussion, it speaks about binding of peptides inducing conformational changes to distant helices. Again, the changes are tiny.

The chemical shifts upon formation of hIRE1 cLD oligomers in Figure 6, especially the chemical shift of Ile128 is significant and large (please see quantifications in Figure 6—figure supplement 2), suggesting a change in the environment of Ile128 due to a conformational change.

While we observe a smaller chemical shift for Ile263 peak upon peptide binding, this peak shifts reproducibly in various experiments where we monitor binding of different peptides and unfolded proteins. Moreover, our XL-MS data where we identified XL-site on the neighboring lysine Lys265 substantiates the NMR data.

2) It seems that the field of ER stress seeks to have one or the other model of regulation of the unfolded protein response win out. Why is it not possible that Ire1 responds both to BiP concentration and unfolded protein load?

The current manuscript focuses on the unfolded protein induced activation of mammalian IRE1 providing evidence that BiP is not the sole/primary regulator of the response in higher eukaryotes. We and others showed clearly for the yeast Ire1 that the interaction with the BiP ortholog Kar2 is dispensable for Ire1 activation and instead it modulates deactivation dynamics of Ire1. In the absence of evidence to the contrary, we remain that it is most plausible that a conserved mechanism is in place for mammalian IRE1 to tune the response. We added discussion to clarify our point, please see the Discussion section, third paragraph.

3) The binding studies conducted by the authors involved peptides. Can the authors comment on binding of intact proteins as well?

We already showed binding of intact CH1 domain of IgG and Staphylococcus nuclease mutant D131D as model unfolded proteins in this study, please see Figure 2, Figure 3 and Figure 2—figure supplement 2 and subsection “hIRE1α cLD binds to unfolded proteins”.

4) A number of concerns were raised concerning the identification of human Ire1 binding sequences. There were no sequences in the MPZ lumenal domain that interacted with recombinant Ire1α-cLD, whereas the sequence that was used for the bulk of experiments is located in the cytosolic domain. Although this doesn't interfere with the conclusion that human Ire can bind peptides, the conclusions about the type of sequences that it can bind may not be reliable, since this protein does induce a UPR in mammalian cells. A second point involves the Sars-CoV, 8ab protein. The paper cited for this protein demonstrates that it binds to ATF6 and induces its cleavage but has no effect on Ire1 or PERK activation. So if a peptide from this protein can bind to Ire1, it is not sufficient to activate it. Lastly, the CH1 domain produced by Marcinowski et al. was oxidized, which resulted in BiP binding to it with BiP's lid in a much more open conformation than when peptides bound to it. Do the authors find that the perturbation of Ire1 luminal domain is much greater in order to accommodate this client than when peptides are used in the NMR studies?

The tiling arrays for the MPZ protein show that both IRE1 cLD and BiP also binds peptides derived from its lumenal domain, albeit less strongly (Figure 2). Yet, even though peptide arrays are powerful in identifying the binding motifs in the context of short peptide sequences, they might not be predictive of interactions in the context of a 3D structure or physiological membrane topology. Therefore it is important to view these results in their intended context, namely as a tool to identify the peptide binding preferences of hIRE1α cLD and BiP, rather than identifying physiologically meaningful interaction within these model proteins. We modified the text accordingly to clarify this point, please see subsection “The lumenal domain of human IRE1α binds unfolded polypeptides”.

We purified CH1 under reducing conditions according to the protocol described in Feige et al. where they showed the binding of BiP to the reduced form of CH1 is slightly stronger than that of the oxidized form. We now clarified this in the Materials and methods section. We do not see the perturbations to be greater upon CH1 binding compared to peptides, indicating that an exposed portion of CH1 contacts hIRE1 cLD in a similar way as our peptide surrogates.